# Sustained microglial depletion with CSF1R inhibitor impairs parenchymal plaque development in an Alzheimer's disease model

Elizabeth Spangenberg[1], Paul L. Severson[2], Lindsay A. Hohsfield[1], Joshua Crapser[1], Jiazhong Zhang[2], Elizabeth A. Burton[2], Ying Zhang[2], Wayne Spevak[2], Jack Lin[2], Nicole Y. Phan[1], Gaston Habets[2], Andrey Rymar[2], Garson Tsang[2], Jason Walters[2], Marika Nespi[2], Parmveer Singh[2], Stephanie Broome[2], Prabha Ibrahim[2], Chao Zhang[2], Gideon Bollag[2], Brian L. West [2] & Kim N. Green[1]

Many risk genes for the development of Alzheimer's disease (AD) are exclusively or highly expressed in myeloid cells. Microglia are dependent on colony-stimulating factor 1 receptor (CSF1R) signaling for their survival. We designed and synthesized a highly selective brain-penetrant CSF1R inhibitor (PLX5622) allowing for extended and specific microglial elimination, preceding and during pathology development. We find that in the 5xFAD mouse model of AD, plaques fail to form in the parenchymal space following microglial depletion, except in areas containing surviving microglia. Instead, Aβ deposits in cortical blood vessels reminiscent of cerebral amyloid angiopathy. Altered gene expression in the 5xFAD hippocampus is also reversed by the absence of microglia. Transcriptional analyses of the residual plaque-forming microglia show they exhibit a disease-associated microglia profile. Collectively, we describe the structure, formulation, and efficacy of PLX5622, which allows for sustained microglial depletion and identify roles of microglia in initiating plaque pathogenesis.

[1] Department of Neurobiology and Behavior, University of California Irvine (UCI), Irvine, CA 92697, USA. [2] Plexxikon Inc, Berkeley, CA 94710, USA. Correspondence and requests for materials should be addressed to K.N.G. (email: kngreen@uci.edu)

Alzheimer's disease (AD) is a progressive, age-related neurodegenerative disorder thought to be triggered by the appearance and build-up of amyloid-β (Aβ) plaques in the cortex[1,2]. These plaques subsequently spread throughout the forebrain and lead to a cascade of events, culminating in synaptic and neuronal loss that underlie the disease-associated memory impairments. Genome-wide association studies have identified numerous genes that confer increased risk for developing the disease; however, the mechanisms underlying plaque formation remain unclear. Discerning commonalities in the function of these disease-associated genes may elucidate potential biological mechanisms involved in the production of plaques[3]. Several of the top identified risk-conveying genes are highly enriched in myeloid cells (*CR1*, *CD33*, *ABCA7*, *TREM2*, *MS4A*, *EPHA1*, *SPI1*[4–6]), highlighting the link between myeloid biology and the risk for developing AD.

Within the CNS, microglia perform homeostatic maintenance, immune-related, and phagocytic functions. Their reported capacity for Aβ phagocytosis and clearance led to the suggestion that age-related changes in microglial function reduce clearance of neuronally derived Aβ from the brain[7], thus allowing plaque formation, as modeled in *ex-vivo* systems[8,9]. Indeed, we and other groups report that following the initial period of plaque formation, microglia surround the plaques and subsequently mount a harmful and non-resolving inflammatory response. Despite this response, however, Aβ clearance and plaque modulation/dynamics is unaffected[10–12], yet the removal of the microglia at advanced stages of pathology protects against synaptic and neuronal loss[10].

Here, we set out to explore the contribution(s) of microglia to plaque formation in the initial stages of the disease, which requires prolonged depletion of microglia throughout the plaque-forming period. To that end, we designed, synthesized, and optimized a potent, specific, orally bioavailable, and brain-penetrant CSF1R inhibitor, PLX5622, to deplete microglia for > 6 months in 5xFAD mice. With the elimination of microglia, we uncovered critical roles of these cells in plaque formation, compaction, and growth, mitigating neuritic dystrophy, and modulating hippocampal neuronal gene expression in response to Aβ pathology. These results implicate microglia as critical and causative in the development and progression of multiple facets of AD pathology.

## Results

**Microglia contain Aβ aggregates in AD mouse models and humans**. The aggregation of Aβ is an initial step in the formation of plaques, requiring an acidic pH[13] and micromolar concentrations of Aβ monomers[14]. The extracellular space does not meet these requisite conditions, suggesting that Aβ aggregation originates elsewhere. In contrast to the extracellular space, microglial lysosomes provide a suitable environment to facilitate Aβ aggregation, potentially contributing to the onset of plaque pathology[15]. To investigate the potential for microglia-mediated plaque formation, we examined Aβ aggregates within microglia in transgenic mouse models of AD. In 15-month-old 3xTg-AD mice, a time point at which plaques are beginning to form, we stained tissue for Thio-S (aggregated Aβ), microglia, and lysosomes. While plaque-associated microglia show accumulation of aggregates within their lysosomes (Fig. 1a, b; as well established[16]), we also observed non-plaque-associated microglia, including ramified microglia, accumulating aggregates within lysosomal compartments (Fig. 1c, d), and microglia containing intracellular aggregates the size of small plaques (Fig. 1e, f). The absence of nearby plaques suggests that existing aggregated Aβ was not the source of these intracellular deposits. Similarly,

non-plaque-associated microglia in 4- and 7-month old 5xFAD brains also showed accumulation of Thio-S+ material (Fig. 1g–j). Additionally, utilizing aged human postmortem brains, including non-demented, high pathology non-demented, and Alzheimer's disease subjects, we again found non-plaque-associated microglia containing Aβ aggregates (Supplementary Figure 1A–I). Thus, microglia in plaque forming regions in mouse and human brains can accumulate aggregated Aβ intracellularly, and we hypothesize that this could be an initial and crucial step toward plaque formation.

**Development of a CSF1R inhibitor for microglial elimination**. To explore the roles of microglia in the initial development of plaque pathology, we required a method that allowed for the extended and specific elimination of microglia throughout the plaque forming period, using non-invasive/non-stressful approaches. As we previously demonstrated, microglia are critically dependent on CSF1R signaling for their survival[17], and we set out to develop specific CSF1R inhibitors that were orally bioavailable, brain-penetrant, and able to achieve robust brain-wide microglia elimination for extended periods of time. Using a structure-guided drug design strategy based on the existing CSF1R/KIT/FLT3 inhibitor PLX3397[10,18], combined with in vivo screening for optimal PK, brain penetrance, and microglial depletion, we created PLX5622 (Fig. 2a), with crystallography revealing the interactions between PLX5622 and the CSF1R (Fig. 2b). The synthesis schematic for the synthesis of PLX5622 and the formation of the optimized PLX5622-fumaric acid salt is shown (Fig. 2c). Cell free kinase inhibitor profiling revealed that PLX5622 is highly specific for the CSF1R, showing > 20-fold selectivity over KIT and FLT3, the two most homologous receptors (Supplementary Table 2, 3). Two key structural differences between PLX5622 and PLX3397 contribute to the improved selectivity based on crystallographic analysis (Fig. 2b; Supplementary Table 4). First, the 2-fluoro substitution on the middle pyridine ring of PLX5622 is designed to access the CSF1R-unique space next to Gly-795 (a gate-keeper to the interior allosteric pocket); both KIT and FLT3 have a bulkier cysteine at the equivalent position. While the difference between a hydrogen atom and a fluorine atom may seem small (with slightly longer bond length and slightly larger radius, a fluorine atom extends the van der Waals edge by ~ 0.5 Å compared to a hydrogen atom), the effect is significant. Anchored on both ends by hydrogen bonds and hydrophobic packing (Fig. 2b), PLX5622 has the fluorine atom at a fixed position. As van der Waals repulsion increases in proportion to the 6th power of the distance between two atoms, a larger gate keeper residue (as seen in KIT and FLT3) will likely incur an energetic penalty for the steric hindrance. In support of this mechanism, a close analog of PLX3397 containing the same 2-fluoro substitution as PLX5622 has the same potency as PLX3397 in inhibiting CSF1R but is 10-fold more selective over KIT than PLX3397. Second, the terminal pyridine group of PLX5622 is optimized to stabilize the allosteric pocket of CSF1R vacated by the displaced juxtamembrane domain. When PLX5622 binds to KIT or FLT3, the position and orientation of the middle pyridine ring causes steric clash with the gate-keeper cysteine, compromising the optimal fit of the terminal pyridine moiety.

In vivo PLX5622 demonstrated desirable PK properties in mice, rats, dogs, and monkeys (Supplementary Table 5), with a brain penetrance of ~20% (compared to ~5% for PLX3397[17]; Supplementary Table 6). The improved blood-brain barrier (BBB) penetrance of PLX5622 over PLX3397 is consistent with the physicochemical properties of the two compounds (Supplementary Table 7). PLX5622 has lower molecular weight, higher

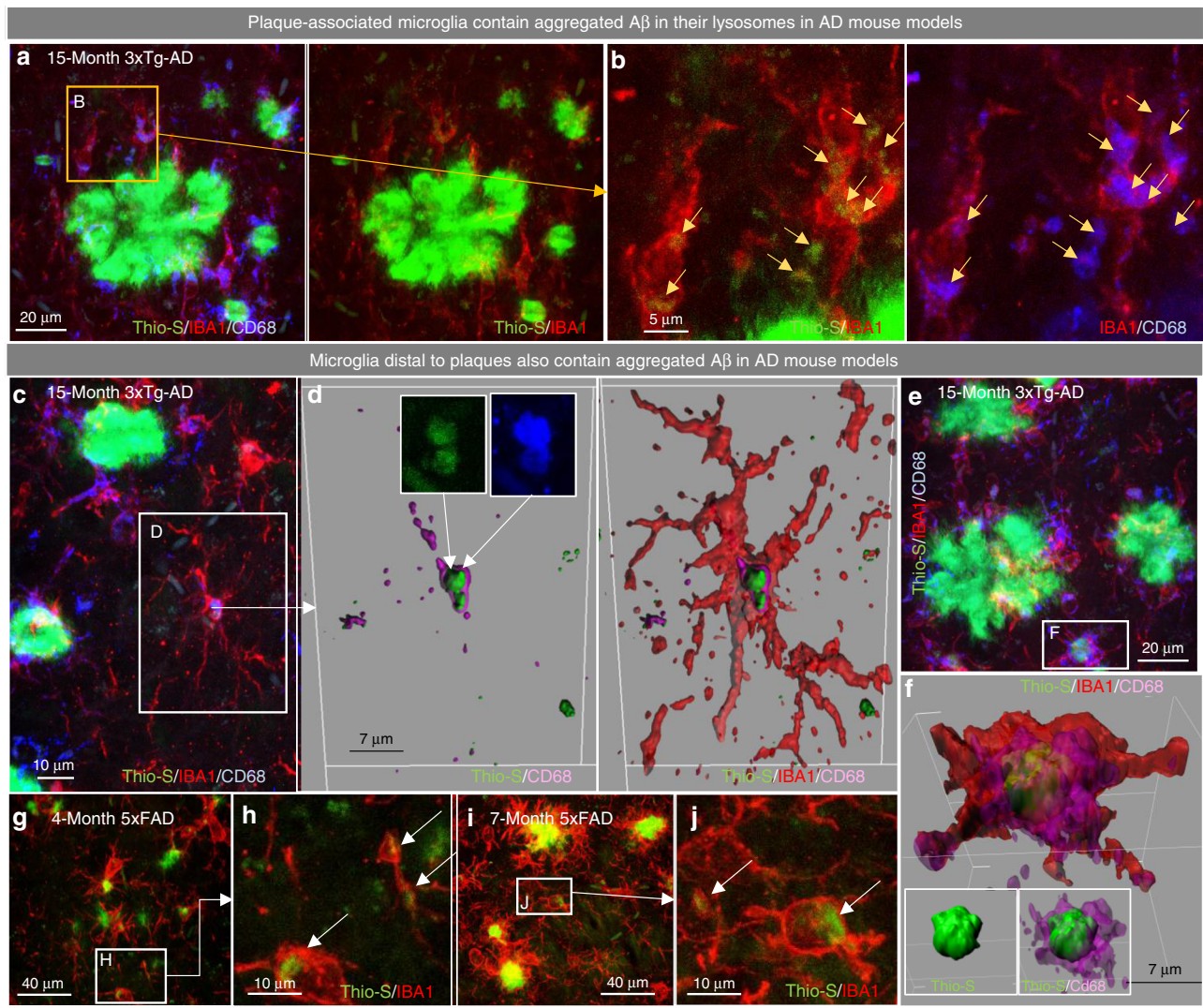

**Fig. 1** Plaque-distal microglia contain aggregated Aβ. **a–e** 15-month-old 3xTg-AD mice were stained for dense core deposits with Thio-S (in green), and immunolabeled for microglia (IBA1 in red) and macrophage lysosomes (CD68 in blue; **a**, **c**, and **e**) with zoomed image (**b**) of Thio-S+ material within microglia and within lysosomes, separately. Scale bars = 20 μm for **a**, **e** 5 μm for **b**, 10 μm for **c**. **d**, **f** Three-dimensional reconstruction of microglia (IBA1 in red), the microglial lysosome (CD68 in purple), and fibrillar Aβ (Thio-S in green), demonstrating the localization of Aβ to the microglial lysosome in non-plaque associated microglia. Scale bars = 7 μm. **g–j** 5xFAD animals stained for dense-core deposits (Thio-S in green) and immunolabeled for microglia (IBA1 in red; **g** and **i**), with zoomed images (**h**, **j**) demonstrating Thio-S+ aggregates in microglial cell bodies in 4- and 7-month-old 5xFAD mice. Scale bars = 40 μm for **g**, **i** 10 μm for **h**, **j**.

lipophilicity and better cell permeability, all factors known to influence the ability of a compound to cross the BBB. PLX5622 was formulated in rodent chow, and administration to mice showed highly effective, causing a 90% reduction with 1200 ppm chow within 5 days of treatment (Fig. 2d, e). Instructions for the synthesis and formulation of PLX5622 are provided (Supplementary Methods).

**Absence of abnormalities with microglial elimination.** Having designed and optimized PLX5622 for microglia depletion, we sought to explore the contributions of these cells in the initial stages of AD pathology. To that end, 5xFAD animals, which exhibit plaque pathology from 3 months of age, underwent treatment at 1.5 months of age with PLX5622-formulated chow (1200 ppm) or control diet continuously for either 10–24 weeks. Four treatment groups were included: Wild-type, PLX5622,

5xFAD, and 5xFAD + PLX5622 (n = 12/group; sex-balanced; Fig. 3a) and initial characterization focused on the 24-week treated animals. Terminal PK values revealed no differences between PLX5622-treated groups in either plasma or brain (Fig. 3b) and no significant differences were seen in circulating leukocyte subsets (Fig. 3d; Gating strategy in Supplementary Figure 2) with treatment.

In the brain, 5xFAD mice displayed increased numbers of microglia/myeloid cells at both 4 and 7 months of age compared to wild-type mice, with masses of enlarged, plaque-associated cells seen throughout the brain (Fig. 3c). Oral PLX5622 treatment led to almost complete microglial elimination (97–100% reduction; Fig. 3c and quantified in e and f), even with 24 weeks of treatment, showing that PLX5622 allows for extended elimination of microglia. The absence of microglia for 24 weeks in wild-type mice did not induce any measurable negative effects in animal behavior, including tests of anxiety (elevated plus maze;

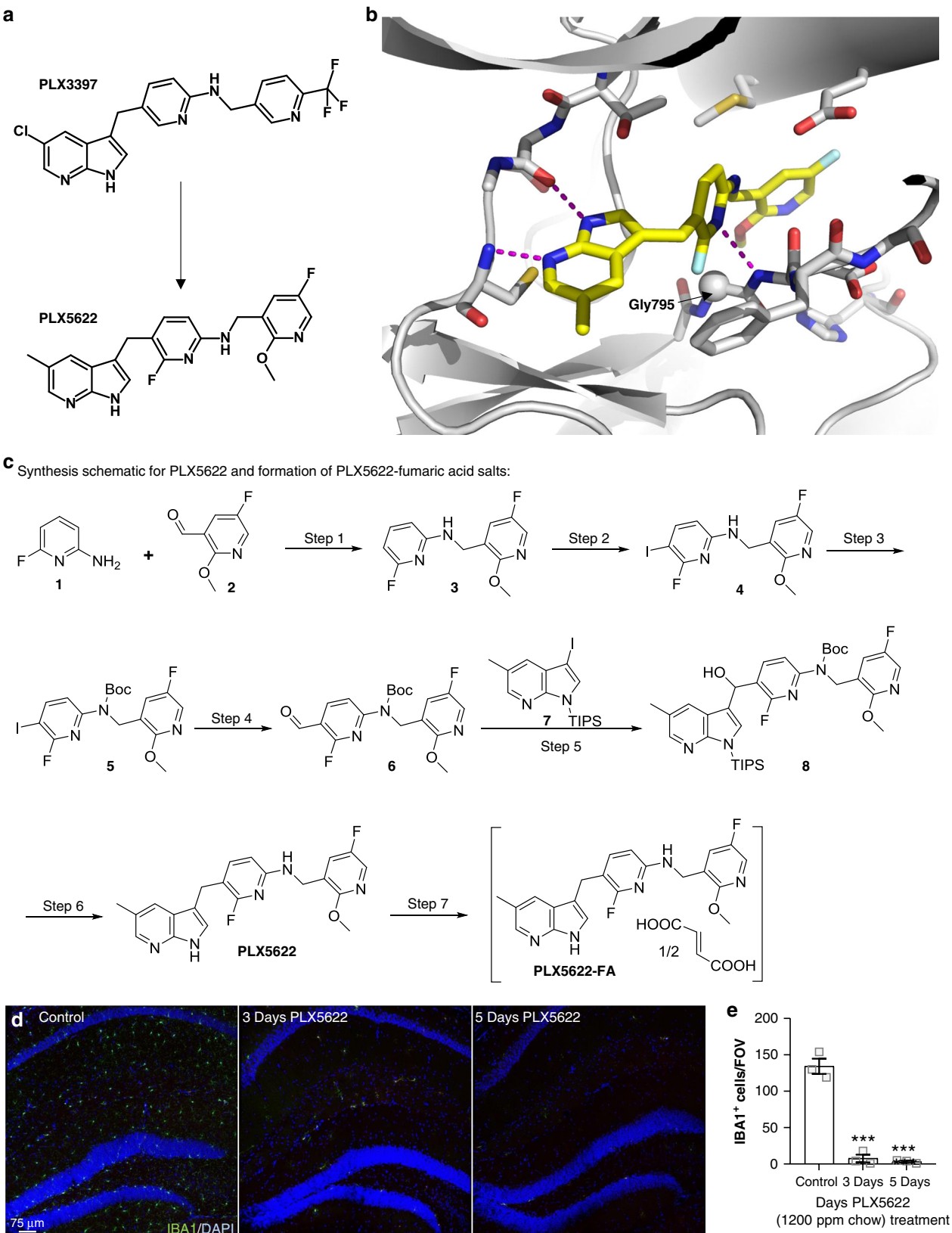

Fig. 3g, h) and motor-function (open field; Supplementary Figure 3A–D). Compared to wild-type animals, 5xFAD mice showed reduced anxiety-related behaviors compared to controls in the elevated plus maze (EPM; Fig. 3g, h). We previously

reported reduced EPM anxiety-related behavior in a model of hippocampal neuronal lesion[18], suggesting that hippocampal functioning in 5xFAD animals at this age may be impaired. With microglial elimination, these behaviors were further exacerbated

**Fig. 2** Design and synthesis schematic of the CSF1R inhibitor PLX5622 for extended microglial elimination. **a** Chemical structure of selective CSF1R inhibitor PLX5622. PLX5622 is structurally similar to CSF1R/KIT/FLT3 inhibitor PLX3397 with modifications concentrated on the two pyridine moieties. **b** X-ray crystal structure of the CSF1R-PLX5622 complex. PLX5622 is anchored to the active site of CSF1R by 3 hydrogen bonds (indicated by dashed lines). 7-Azaindole of PLX5622 forms two hydrogen bonds with the hinge region of CSF1R whereas the central pyridine ring forms a hydrogen bond with the main chain amino group of Phe796. The CSF1R selectivity is largely determined by the interaction between PLX5622 and Gly795 (represented as a sphere), which is a bulkier residue (cysteine) in KIT and FLT3. The substitutions on the tail pyridine ring also contribute to the selectivity. This group displaces the juxatemembrane (JM) region (absent in the structure) from the allosteric site. In contrast, PLX3397 binds CSF1R in the presence of the JM region. **c** Synthesis schematic for PLX5622 and subsequent formation of PLX5622-fumaric acid salt. PLX5622 was synthesized from commercially available 2-amino-6-fluoropyridine (**1**), 5-fluoro-2-methoxypyridine-3-carbaldehyde (**2**), and 3-iodo-5-methyl-1-(triisopropylsilyl)-1H-pyrrolo[2,3-b]pyridine (**7**) using the reaction scheme. **d** Immunolabeling of microglia (IBA1 in green) and cell nuclei (DAPI in blue) of two-month-old animals treated with control or 1200 ppm PLX5622-formulated chow. Scale bar = 75 μm. **E**, Quantification of hippocampal microglial number over 5 days of treatment with PLX5622 ($p < 0.001$ for 3d and 5d). Two-way ANOVA with Dunnet's post hoc test. $n = 3$ for Control, $n = 3$ for 3d PLX5622, $n = 3$ for 5d PLX5622. Statistical significance is denoted by ***$p < 0.001$. Error bars indicate SEM

(Fig. 3g, h). To assess hippocampal-dependent memory, mice underwent testing using Y-Maze, Morris water maze (MWM), and Contextual fear conditioning (CFC) and 5xFAD animals did not show impairments in any of these tasks (Supplementary Figure 3E–G; Fig. 3i–k). In the MWM probe trial, PLX5622-treated wild-type mice tended to spend more time in the platform zone compared to wild-type mice (Fig. 3k), indicating that the long-term absence of microglia is not detrimental to murine cognitive function and may be beneficial, at least in this assay, which is consistent with our prior findings[17,18].

**Microglial elimination impairs plaque formation.** Having demonstrated robust microglial elimination prior to and throughout the period of plaque formation in 5xFAD mice, we next sought to determine the consequences of microglial ablation on pathology. In wild-type mice, CSF1R inhibitor treatment for 10 weeks eliminated > 99% of microglia in the cortex (Fig. 4a–f), but a fraction of cells remained in the thalamus (Fig. 4i; enlarged in Supplementary Figure 4). Treatment of 5xFAD mice with PLX5622 again reduced > 99% of microglia in the cortex (Fig. 4d, g), but as noted in wild-type mice, a fraction of cells remained in the thalamus (Fig. 4j). In the absence of microglia, we observed a stark lack of dense-core plaques within cortical regions (quantified in Fig. 4l; retrosplenial (RS) and somatosensory (SS) cortices examined). Notably, a band of plaques were found in the RS cortex, but these were primarily associated with microglia that had survived treatment (Fig. 4d). Plaques were also formed within the thalamus but were again predominantly associated with surviving microglia (Fig. 4j). Close examination of the Thioflavin-S (Thio-S) stained slices revealed deposits beginning to form within cortical blood vessels (Fig. 4d, g – yellow arrows). These results suggest that microglia may be critical regulators of plaque formation and that few surviving microglial cells are sufficient to facilitate some degree of plaque formation (*i.e.*, as seen in the thalamus).

Consistent with the 10-week treated 5xFAD mice, examination of pathology in the 24 week treated mice revealed diminished Thio-S$^+$ dense-core plaque numbers in the cortices of the 5xFAD mice devoid of microglia (Fig. 4e, h; quantified in M). In the 10-week treated cohort, we noted the appearance of plaques in areas that exhibited small populations of surviving microglia (RS cortex and thalamus; Fig. 4d, j). With an additional 14 weeks of treatment (*i.e.*, 24 weeks treated animals), microglia were eliminated brain wide (Fig. 4e, k), but Thio-S plaques persisted in these areas (*i.e.*, the thalamus and RS cortex). Average plaque volumes were ~30–40% smaller, across many brain regions (Fig. 4n), suggesting that microglia contribute to plaque growth in the 5xFAD brain. Thus, even with increased treatment duration

to 7 months of age, plaque formation is prevented in the absence of microglia.

Astrocytes surrounding plaques in the cortex of 7-month-old 5xFAD mice appeared activated, as assessed by GFAP immunoreactivity, which usually is absent in the cortex of non-diseased brains (Supplementary Figure 5A–B). In contrast, GFAP expression was sharply reduced in 5xFAD mice devoid of microglia (Supplementary Figure 5A–C). Absolute numbers of astrocytes were not highly affected by pathology or microglial depletion, as measured by S100β$^+$ cells (quantified in Supplementary Figure 5D), except for a small increase in the RS cortex in 5xFAD mice. These data suggest that microglia regulate astrocyte reactivity in disease, fully consistent with previous reports[19,20].

Notably, abundant Thio-S staining was discovered within large cortical blood vessels in the 5xFAD mice devoid of microglia (*i.e.*, Fig. 4e, h), indicative of cerebral amyloid angiopathy (CAA). Hemisphere stitches of Thio-S staining show clear vascular pathology in microglia-depleted 5xFAD mice (Fig. 4b). The appearance of CAA occurred only in microglia-depleted 5xFAD mice, as CAA was not observed in untreated 5xFAD mice or in treated wild-type mice. In addition to Thio-S, CAA and plaques stained postively for Aβ fibrils, via the conformation specific antibody OC[21] (Fig. 4r; Supplementary Figure 6C–D) and Aβ42 (Fig. 4s; Supplementary Figure 6E–F), but were negative for Aβ oligomers via A11 antibody[22] (Fig. 4q; Supplementary Figure 6A–B). Staining for pyroglutamate-3 Aβ, which forms the plaque cores[23], revealed a sharp reduction in immunoreactivity in PLX5622-treated 5xFAD mice in the cortex (Fig. 4p; Supplementary Figure 6G–H), along with reductions in 6E10$^+$ diffuse plaques (Fig. 4o; Supplementary Figure 6G–H). Of note, pyroglutamate-3 immunoreactivity was found in areas of treated 5xFAD mice that exhibited plaque pathology (*i.e.*, thalamus; Supplementary Figure 6I–J).

As blood vessel accumulation of Aβ is known to increase the risk of intracranial hemorrhage and hemorrhagic stroke in human brains[24], we wanted to evaluate whether similar pathology was developing in microglia-depleted 5xFAD animals. Examination for microbleeds in 4- (Supplementary Figure 7A–D) and 7- (Supplementary Figure 7E–H) month cohorts of mice revealed notable iron deposits in the thalamus of four treated 5xFAD mice (2 animals/cohort). While surprising to find strong thalamic microbleeds in a subset of treated 5xFAD mice, transgenic AD animals that develop primarily CAA and exhibit extensive Aβ deposition in thalamic microvasculature also show an increase in thalamic microbleeds[25], highlighting the susceptibility of this region to develop vascular-related impairments/dysfunction. Immunolabeling for the endothelial tight junction marker, Claudin-5, revealed an increase in Thio-S positive blood vessel diameter

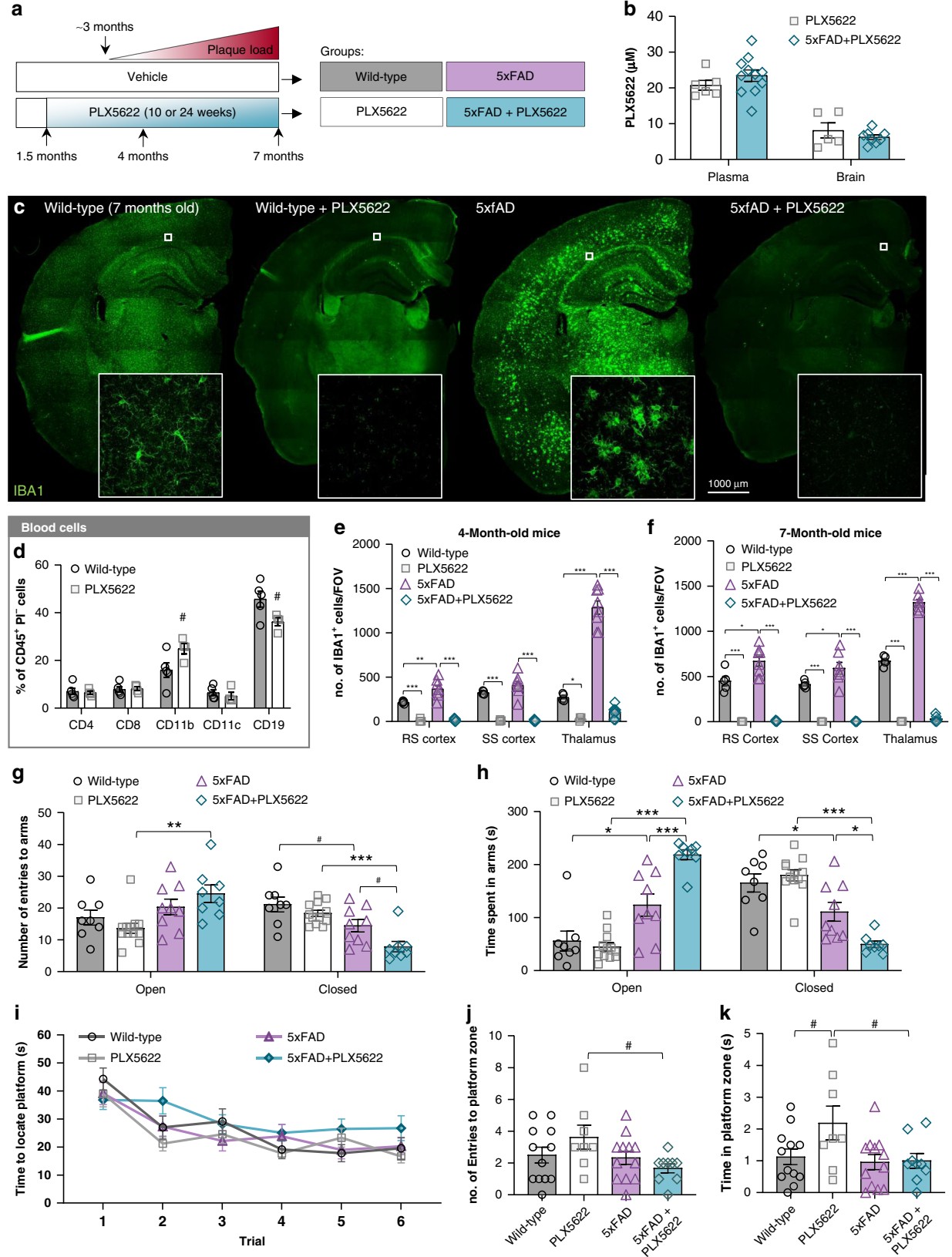

(Supplementary Figure 7I–K), consistent with prior data[26], and reduced intensity of Claudin-5 staining in Aβ-associated blood vessels (Supplementary Figure 7L), suggesting impaired vascular integrity in the presence of Aβ pathology.

The absence of microglia did not significantly alter the levels of Aβ38, Aβ40, or Aβ42, in detergent-soluble or -insoluble cortical and thalamic homogenates in either the 4- or 7- month cohorts (Fig. 5a–h), in line with multiple studies showing that CSF1R

**Fig. 3** Extended elimination of microglia does not induce peripheral leukocyte or behavioral abnormalities. 1.5-month-old wild-type (WT) or 5xFAD mice were treated with control chow or PLX5622 for 10 or 24 weeks. **a** Experimental design. **b** Terminal PK of PLX5622 and 5xFAD + PLX5622 groups. Two-tailed independent t-test. $n = 5–6$ for PLX5622, $n = 8–11$ for 5xFAD + PLX5622. **c** Hemisphere stitches of microglia (IBA1 in green) in the 7-month-old cohort. Scale bar = 1000 μm. **d** Analysis of different subsets of leukocytes (CD11b, $p = 0.059$; CD19, $p = 0.052$; all others NS). Two-tailed independent t-test; $n = 4–5$ for Wild-type, $n = 3–4$ for PLX5622. **e–f** All analyses listed in respective order for retrosplenial (RS) cortex, somatosensory (SS) cortex, and thalamus. Microglial number in 4 **e** and 7 **f** month-old cohorts (4-month cohort: WT v 5xFAD, $p = 0.001$, NS, $p < 0.001$; WT v PLX5622, $p < 0.001$, $p < 0.001$, $p = 0.044$; 5xFAD v 5xFAD + PLX5622, $p < 0.001$, $p < 0.001$, $p < 0.001$. 7-month cohort: WT v 5xFAD, $p = 0.015$, $p = 0.024$, $p < 0.001$; WT v PLX5622, $p < 0.001$ for all regions; 5xFAD v 5xFAD + PLX5622, $p < 0.001$ for all regions). Two-way ANOVA with Tukey's post hoc test; $n = 4–6$ for Wild-type, $n = 4–6$ for PLX5622, $n = 5–8$ for 5xFAD, $n = 6–9$ for 5xFAD + PLX5622. **g–h** Measurements for anxiety were performed by Elevated Plus Maze (EPM) and quantified as number of arm entries (**g** WT v 5xFAD, NS (open) and $p = 0.050$ (closed); PLX5622 v 5xFAD + PLX5622, $p = 0.007$ (open) and $p < 0.001$ (closed), 5xFAD v 5xFAD + PL5622, NS (open) and $p = 0.056$ (closed)) and time in arms (**h** WT v 5xFAD, $p = 0.0157$ (open) and $p = 0.046$ (closed); PLX5622 v 5xFAD + PLX5622, $p < 0.001$ (open) and $p < 0.001$ (closed); 5xFAD v 5xFAD + PLX5622, $p < 0.001$ (open), $p < 0.020$ (closed)). **i** Mean latencies to a hidden platform from the Morris water maze (MWM) acquisition trials. **j–k** Number of platform entries (**j**; PLX5622 v 5xFAD + PLX5622, $p = 0.071$) and time in platform zone (**k**; WT v PLX5622, $p = 0.094$; PLX5622 v 5xFAD + PLX5622, $p = 0.073$) in the MWM probe trial. For all behavioral analyses: Two-way ANOVA with Tukey's post hoc test; $n = 8–12$ for Wild-type, $n = 8–12$ for PLX5622, $n = 9–12$ for 5xFAD, $n = 8–9$ for 5xFAD + PLX5622. Statistical significance is denoted by *$p < 0.05$, **$p < 0.01$, and ***$p < 0.001$. Statistical trends are denoted by #$p < 0.10$. NS, not significant ($p > 0.10$). Error bars indicate SEM

inhibition does not affect Aβ production/clearance[10,27,28]. Thus, the absence of microglia modulates the location of Aβ accumulation (*i.e.*, parenchyma vs. vasculature) but does not alter the net amounts present.

Investigations into oligomeric species of Aβ (via A11 immunoblotting) found no significant changes in protein levels with microglia elimination in 5xFAD mice (Fig. 5i, j). To confirm that APP processing was unchanged with microglia elimination in 5xFAD mice at the protein level, we immunoblotted cortical tissue for various components of the APP processing pathway including APP and its cleavage products, as well as proteins associated with α- (ADAM10), β- (BACE1), and γ- (PEN2 and PS1) secretase activity. We found significant elevations in full-length (fl) APP and Carboxy-terminal fragments of APP (C99 and C83) in 5xFAD mice relative to wild-type (Fig. 5k, l), but observed no reductions in protein levels with PLX5622 treatment in 5xFAD animals. Additionally, gene expression analyses of AD-related genes were performed (Fig. 5m; methods described in greater detail for Figs 8, 9) and we found minimal changes with microglia depletion, other than in microglial expressed genes (*i.e.*, *Trem2*, *Spi1*, *Inpp5d*, and *Ctsd*).

**Microglial plaque compaction ameliorates neuritic dystrophy.** Upon examining caudal brain regions, we found some IBA1+ cells present within the subiculum and associated white matter tracts in the otherwise microglia-devoid 5xFAD animals (mice treated through 1.5–7 months of age; Fig. 6a, b). While PLX5622 treatment reduced myeloid cell numbers in the subiculum (77% reduction; Fig. 6e), we observed a tight spatial association between surviving cells and Thio-S plaques (Fig. 6b, d), similar to that seen in the thalamus of the 10-week treated 5xFAD mice (Fig. 4j). In fact, plaques were predominantly observed in the immediate vicinity of surviving IBA1+ cells – within the subiculum and associated white matter tract – while the cortex contained only vascular Thio-S+ deposits, in stark contrast to untreated 5xFAD mice (Fig. 6a vs. b). Quantification of Thio-S+ subicular plaques revealed a 33% decrease in plaque number (Fig. 6f) with PLX5622 treatment in 5xFAD mice. Importantly, these results highlight a clear relationship between microglia/myeloid cells and the appearance of plaques, suggesting a role of microglia/myeloid cells in facilitating plaque formation, and emphasizes that few cells are required for this process to occur.

Structural examination of the plaques that had formed in PLX5622-treated 5xFAD mice (treated through 1.5–7 months of age), presumably due to the presence of surviving microglia at the

time of their formation, revealed morphological differences compared to plaques formed and maintained in the presence of microglia (Fig. 6g–i). These plaques displayed an irregular shape with structured filaments radiating out from a central fissure/point and lacked the homogenous staining intensity of plaques in microglia-intact 5xFAD mice. As such, both plaque circularity (Fig. 6j) and mean Thio-S intensities (Fig. 6k) were reduced in the 5xFAD mice devoid of microglia. These features were common to all areas explored, including in the subiculum, where some IBA1+ cells were still present, suggesting that some threshold density of microglial cells may be required for full plaque compaction/restructuring.

Focusing on plaque structure, we next evaluated the presence of ApoE within plaque cores – previous reports describe the coaggregation of ApoE with Aβ fibrils in both plaques and vascular pathology[29,30]. Consistent with this, ApoE immunoreactivity was apparent within dense core plaques, but also in cellular processes surrounding the cores (Fig. 6l). Colocalization with the microglia/myeloid marker IBA1 revealed that these ApoE positive processes were indeed microglia (Fig. 6l), confirming that plaque-associated microglia are a source of ApoE[30]. Examination of ApoE immunoreactivity in microglia-devoid 5xFAD mice showed that the ApoE+ microglial processes surrounding dense core plaques were absent, as expected, but also the plaque cores themselves lacked ApoE staining (Fig. 6m). We quantified ApoE immunoreactivity in Thio-S dense core plaques in the cortex and thalamus, revealing a ~50–70% reduction in core ApoE (Fig. 6n). Thus, microglia surrounding plaques are a major source of plaque-associated ApoE, supporting recent findings[30,31].

Dystrophic neurites are found in a halo around fibrillar plaques and are characterized by the accumulation of LAMP1 and APP[32–34]. To evaluate the role of microglia in regulating the growth of dystrophic neurites, we stained for LAMP1 (Fig. 6o) and APP (Fig. 6q) and found that the absence of microglia enhanced the dystrophic neurite halo area (Fig. 6p, r). These results are consistent with prior studies showing that microglia form a physical barrier around plaques and compact Aβ into dense deposits, protecting against local neurite damage[35–37].

**Alternative CSF1R inhibitor confirms diminished plaque load.** Given the striking effects of PLX5622 treatment/microglial depletion on reducing plaque formation, we wanted to confirm the results with alternative CSF1R inhibitor paradigms. We established that PLX3397 formulated at 600 ppm in chow could

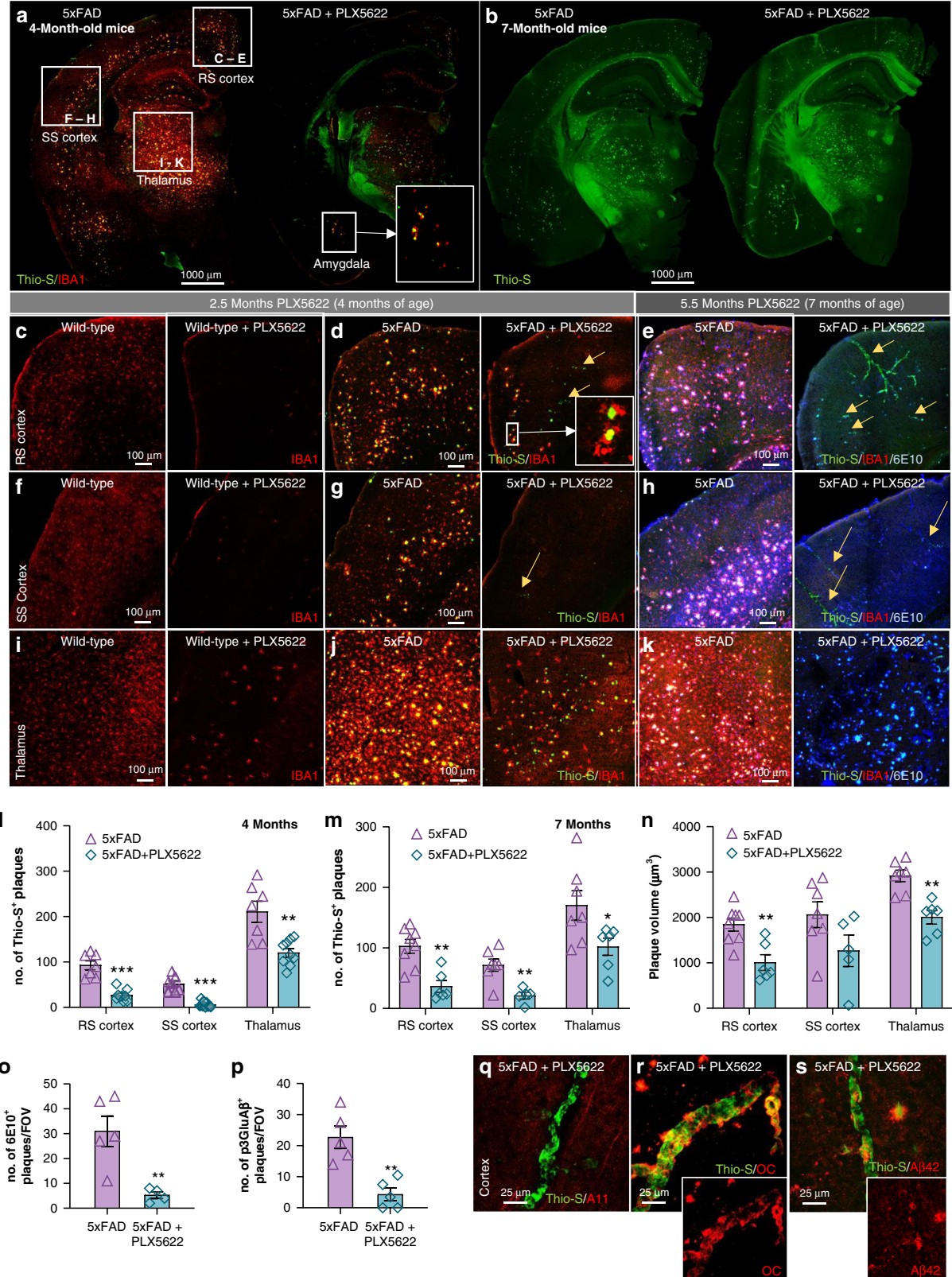

achieve robust brain-wide microglial elimination (>99%)[38], and thus, treated 1.5-month-old 5xFAD mice with this formulation until 5 months of age (Fig. 7a). Again, examination of the brains of these treated mice showed minimal plaque pathology and the appearance of CAA (Fig. 7c, d – zoomed image of CAA in D). We

further examined the role of peripheral CSF1R inhibition on AD pathology by treating 1.5-month-old 5xFAD mice until 5 months of age with PLX3397 (75 ppm) chow (Fig. 7a) – a formulation that provides robust peripheral CSF1R inhibition without microglial elimination[17]. This lower dose achieved brain levels of

**Fig. 4** Long-term elimination of microglia in 5xFAD mice reduces plaque number and volume and is accompanied by cerebral amyloid angiopathy (CAA) onset. All analyses listed in respective order for retrosplenial (RS) cortex, somatosensory (SS) cortex, and thalamus. **a** Representative hemisphere stitches of dense-core deposits (Thioflavin-S (Thio-S) in green) and microglia (IBA1 in red). Scale bar = 1000 μm. **b** Representative images of brain hemispheres stained with Thio-S, illustrating the appearance of cerebral amyloid angiopathy (CAA) throughout the cortex of 5xFAD mice devoid of microglia. Scale bar = 1000 μm. **c–d**, **f**, **g**, **i**, **j** Confocal images of sections from 10 week treated animals stained for dense-core plaques (Thio-S in green) and immunolabeled for microglia (IBA1 in red). Scale bar = 100 μm. **e**, **h**, **k** Images from 24 week treated animals stained for dense-core plaques (Thio-S in green) and immunolabeled for microglia (IBA1 in red) and diffuse plaques (6E10 in blue). Scale bar = 100 μm. **l**, Quantification of Thio-S plaque number in 4-month-old cohort ($p < 0.001$, $p < 0.001$, $p = 0.001$). Two-tailed independent $t$-test; $n = 7$ for 5xFAD, $n = 8$ for 5xFAD + PLX5622. **m** Quantification of Thio-S$^+$ plaque number in 7-month-old mice ($p = 0.001$, $p = 0.004$, $p = 0.041$). Two-tailed independent $t$-test; $n = 7$–8 for 5xFAD, n = 5–6 for 5xFAD + PLX5622. N, Plaque volumes in 7-month-old mice ($p = 0.003$, NS, $p = 0.007$). Two-tailed independent $t$-test; $n = 7$–8 for 5xFAD, $n = 5$–6 for 5xFAD + PLX5622. O-P, Quantification of 6E10 ($p = 0.002$) and pyroglutamate-3-modified Aβ ($p = 0.008$) in the cortex of 7-month-old animals. Two-tailed independent $t$-test; $n = 4$–5 for 5xFAD, $n = 4$–5 for 5xFAD + PLX5622. **q–s**, 7-month-old PLX5622-treated 5xFAD animals stained for dense core deposits (Thio-S in green) and immunolabeled for oligomeric Aβ (A11), protofibrillar Aβ (OC) and Aβ$_{1-42}$, respectively. Scale bar = 25 μm. Statistical significance is denoted by *$p < 0.05$, **$p < 0.01$, and ***$p < 0.001$. NS, not significant ($p > 0.10$). Error bars indicate SEM

~1 μM, compared to ~10 μM with the higher dose (Fig. 7b), and accordingly, microglia numbers were unchanged relative to untreated animals (Fig. 7e–i). With 75 ppm PLX3397, no reductions in plaque load or appearance of CAA were detected in treated 5xFAD animals compared to controls (Fig. 7f, h, j–k). Thus, peripheral CSF1R inhibition or simply the presence of CSF1R inhibitor within the CNS do not attribute to the reductions in plaque pathology and more complete microglial elimination is necessary to perturb plaque formation processes.

**Gene expression analyses across three brain regions.** Given the prolonged absence of microglia throughout the adult life of CSF1R-inhibitor-treated wild-type and 5xFAD mice, the stark reduction in plaque formation in the absence of microglia, and the differences in pathology deposition and microglial number in various brain regions (*i.e.*, no plaques but CAA in cortex, plaques but no microglia in thalamus and hippocampus/subiculum), we explored alterations in regional gene expression (mice treated through 1.5–7 months of age). To that end, brains were microdissected into cortical, hippocampal, and thalamic + striatal regions. RNA was extracted, and subsequently analyzed by RNA-seq ($n = 4$/group; Principle Component Analyses shown in Supplementary Figure 8).

Extensive microgliosis across all brain regions in 5xFAD mice: To assess the global effects of pathology on gene expression, we compiled a list of genes significantly changed (using raw p-values) in all 3 brain regions in 5xFAD vs. wild-type mice (Fig. 8a). All common genes were upregulated and the vast majority are microglia-associated, reflecting the microgliosis occurring in 5xFAD mice. The elevation levels averaged 1-fold-change higher across all homeostatic microglial genes, consistent with the observed increase in microglial numbers (Fig. 3f). Consistent with these genes being predominantly expressed by microglia, their expression was markedly reduced with treatment in either 5xFAD or wild-type mice. We selected a subset of homeostatic microglial genes (*Csf1r*, *Cx3cr1*, *C1qa*, *Hexb*, *Siglech*, and *Spi1*) and displayed the expression (RPKM) values for each of the brain regions (Fig. 8b). The expression of these genes was nearly undetectable from PLX5622-treated mice, highlighting their specific expression by microglia and the specificity of PLX5622 on microglial survival. Of note, the expression of homeostatic microglial genes was higher in the 5xFAD + PLX5622 hippocampus than other microdissected brain regions (or wild-type mice), reflecting the population of surviving microglia identified in the subiculum (Fig. 6d). Next, we sought to characterize the gene expression profile of the surviving, plaque-forming microglia in treated 5xFAD animals. To that end, we searched for genes upregulated in 5xFAD brains that were subsequently downregulated with

PLX5622 treatment (to select for microglia-expressed genes) but not present in wild-type brains. We identified expression of *Ccl6*, *Clec7a*, *Cst7*, *Ctsd*, *Ctsz*, and *Itgax* as main genes that follow this expression pattern (Fig. 8c), as well as *Asb10*, *B2m*, *Ccl3*, *Ch25h*, *Gpr65*, *Grn*, *Hcar2*, *Hexa*, *Ly9*, *Lyz2*, *Oasl2*, *Pdcd1*, *Plcg2*, and *Treml2*. Notably, several of these genes are known markers of disease-associated microglia[39] (DAM). Consistent with our expectations, we detected expression of many of these genes (*i.e.*, *Ccl6*, *Clec7a*, and *Cst7*) in PLX5622-treated 5xFAD hippocampi (but not cortex or thalamus), reflecting the population of surviving microglia in this region (see images of subicula in Fig. 6c–d, which are included in microdissected hippocampi) and providing a gene expression signature of plaque-forming/associated microglia.

Microglial depletion prevents the downregulation of synaptic genes in the hippocampus: We compared the number of differentially expressed genes (adjusted $p$-values < 0.05) induced by AD pathology in each of the three microdissected brain regions (*i.e.*, wild-type vs. 5xFAD) and found that the hippocampus was most impacted (413 genes), while both cortex and thalamus exhibited fewer changes in gene expression (141 and 107 genes respectively; Fig. 9a). Transgene expression in 5xFAD mice (consisting of *App*, *Psen1*, and *Thy1*) was increased by ~2-fold over endogenous expression across brain regions (Fig. 9b) and was not impacted by treatment, consistent with the protein expression data (Fig. 5k, l). Initially focusing on the 413 gene expression changes in the hippocampus induced by pathology, we plotted a heat map of the log(2) fold change for each of the 9 relevant comparisons (5xFAD vs. wild-type, 5xFAD + PLX5622 vs. 5xFAD, and PLX5622 vs. wild-type, each for cortex, hippocampus, and thalamus; Fig. 9c). The hippocampus displayed a unique gene expression profile relative to the cortex and thalamus and included a unique subset of downregulated genes. Of note, only 1 downregulated gene was identified in the cortex and none were detected in the thalamus. Furthermore, all gene expression changes (either upregulated or downregulated) appeared to be fully reversed in the hippocampus of treated 5xFAD mice, with fold-changes in 5xFAD + PLX5622 vs. 5xFAD hippocampus being equal and opposite to those induced in the 5xFAD mice (highlighted by red border). Of the 413 identified genes, 77 were downregulated in the 5xFAD hippocampus (Fig. 9c, and individual genes plotted in D). Notably, these transcripts were primarily synaptic and neuronal in nature, including genes such as *Dlk2*, *Dync1l1*, *Gls*, *Kcnq3*, *Nrg3*, and *Scn1b*. In the absence of microglia, this disease-associated downregulation of neuronal and synaptic genes was prevented (Fig. 9e), highlighting the detrimental role microglia may have on hippocampal plasticity in the 5xFAD brain.

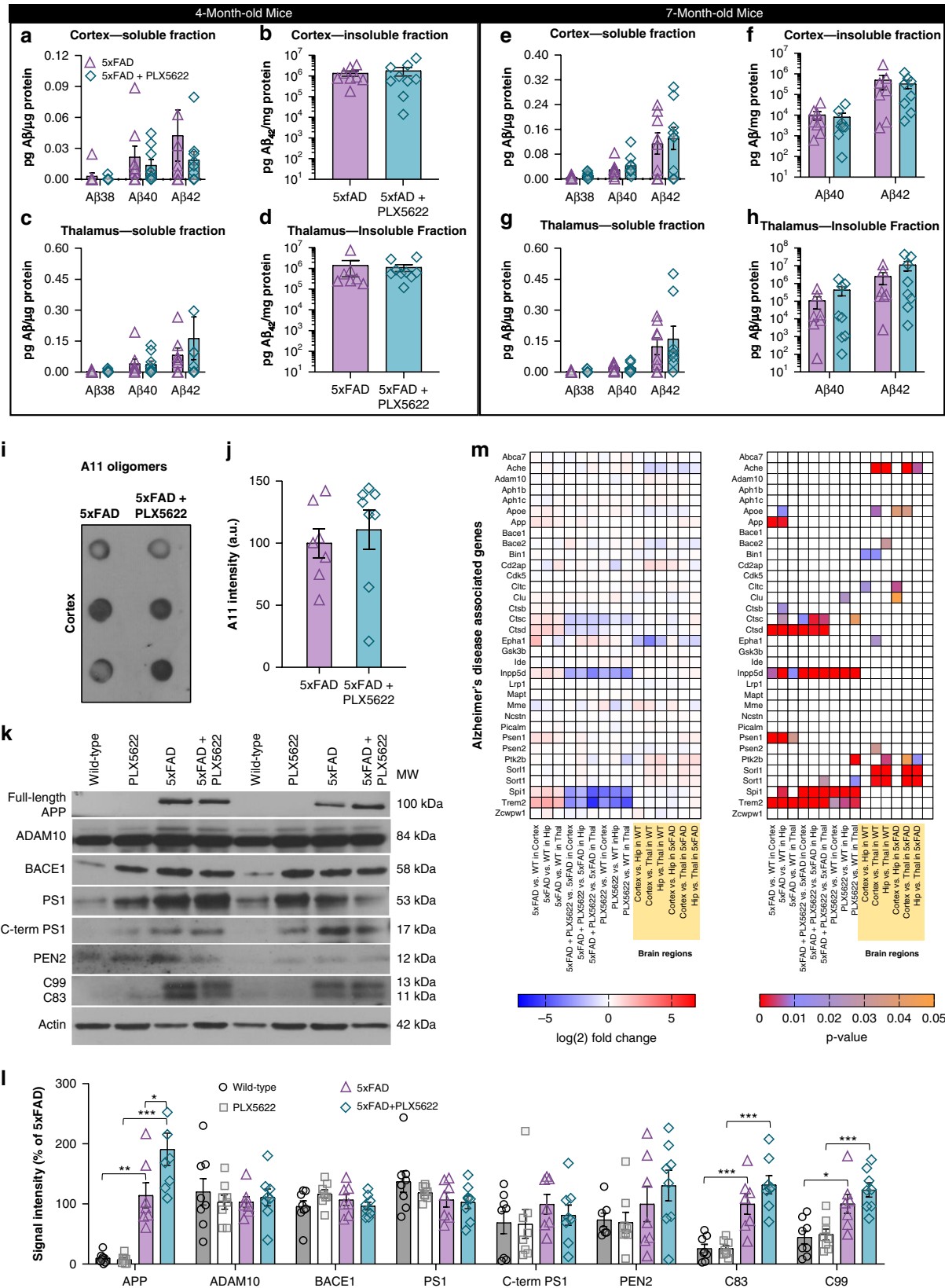

Enrichment analyses of the hippocampus revealed an upregulation in immune function-associated pathways in 5xFAD vs. wild-type, and accordingly, these pathways were all highly significantly downregulated with PLX5622 treatment (Fig. 9f). Analyses of downregulated pathways in 5xFAD vs. wild-type hippocampi revealed that the most highly significant pathways were related to neuronal functioning, such as glutamate receptors, synaptic vesicles, and neuronal membranes (Fig. 9f). Importantly, many

**Fig. 5** No detectable alterations in Aβ levels or APP processing with microglia elimination in 5xFAD mice. **a–h** Aβ levels from cortical (**a–b**, **e–f**) or thalamic brain homogenates (**c–d**, **g–h**) from 5xFAD mice treated with vehicle or PLX5622 (1200 ppm in chow) from 1.5 months of age to either 4 (**a–d**), or 7 (**e–h**) months of age, for both the detergent-soluble and insoluble fractions. In the 4-month-old mice, insoluble $A\beta_{1-38}$ and $A\beta_{1-40}$ were below detection threshold. In the 7-month-old mice, insoluble Aβ levels were plotted on log(10) scale and insoluble $A\beta_{1-38}$ was below detection threshold. Two-tailed independent t-test. For 4-month-old cohort: $n = 7$ for 5xFAD, $n = 9$ for 5xFAD + PLX5622. For 7-month-old cohort: $n = 7$ for 5xFAD, $n = 8$ for 5xFAD + PLX5622. **i–j** Cortical homogenates of 7-month-old mice immunoprobed and quantified, respectively, for A11. Two-tailed independent t-test; $n = 7$ for 5xFAD, n = 8 for 5xFAD + PLX5622. Source data are provided as a Source Data file. **k–l** Levels of components of the amyloid-precursor protein (APP) processing pathway in cortical homogenates from 7-month-old animals (Full length APP: WT v 5xFAD, $p = 0.001$; PLX5622 v 5xFAD v PLX5622, $p < 0.001$; 5xFAD v 5xFAD + PLX5622; $p = 0.020$. C83: WT v 5xFAD, $p < 0.001$; PLX5622 v 5xFAD v PLX5622, $p < 0.001$. C99: WT v 5xFAD, $p = 0.015$; PLX5622 v 5xFAD v PLX5622 $p < 0.001$). Two-way ANOVA with Tukey's post hoc test; $n = 8$ for Wild-type, $n = 8$ for PLX5622, $n = 8$ for 5xFAD, $n = 8$ for 5xFAD + PLX5622. Source data are provided as a Source Data file. **M**, Left panel - heatmap of log(2) fold change of genes associated with AD, including APP/Aβ production and metabolism shown for each of the 9 comparisons and the 6 comparisons between brain regions. Right panel – heatmap of the corresponding p-values for each of the comparisons. Log(2) fold change and p-values indicated by respective scale bar. $n = 4$ for all groups. Error bars indicate SEM

of these pathways were upregulated – or more accurately, never downregulated – with the absence of microglia (Fig. 9f), indicating that microglia influence neuronal gene expression pathways in response to AD pathology. To confirm these gene expression results, we measured gene expression levels from wild-type, PLX5622, 5xFAD, and 5xFAD + PLX5622 hippocampal RNA using a Nanostring Neuropathology panel. Accordingly, overlapping microglia genes were increased and neuronal genes decreased in 5xFAD mice compared to wild-type mice, which did not occur with PLX5622 treatment (Supplementary Figure 9).

Neither modulation of AD pathology nor microglial number grossly alter AD-related gene expression: Although we confirmed that the absence of microglia/treatment with PLX5622 did not alter the amount of Aβ produced, we wanted to explore the impact of pathology and microglia on genes associated with APP processing, Aβ clearance and metabolism, and AD in general (Fig. 5m). The only significant changes in gene expression in 5xFAD mice compared to wild-type were the upregulation of the transgenes (*App* and *Psen1;* as expected due to overexpression), and *Apoe*, all of which were not altered by the absence of microglia, and the downregulation of myeloid-expressed *Trem2*, *Ctsb*, *Ctsc*, and *Ctsd* in the absence of microglia, as expected. Comparisons of changes in expression between brain regions showed regional differences in *Ache*, *Apoe*, *Ptk2b*, *Sorl1*, and *Sort1*. Thus, both AD pathology and the absence of microglia have minimal effects on AD-related genes, and importantly, we observe no alterations in gene expression or protein production in the absence of microglia that could account for reduced plaque formation.

**Microglial repopulation seeds plaques.** While microglia can be indefinitely depleted via the continued administration of CSF1R inhibitors, the microglial compartment can also be repopulated upon CSF1R inhibitor withdrawal[17,40,41]. To further prove that microglia are responsible for plaque formation, we sought to examine the effects of microglial repopulation in 5xFAD mice after 10 weeks of PLX5622 treatment. We treated a cohort of 1.5-month-old 5xFAD mice with PLX5622 (1200 ppm in chow) to eliminate microglia until 4 months of age, then CSF1R inhibitor-formulated chow was removed to stimulate microglial repopulation and the brains were examined one month later (Fig. 10a). Microglia repopulated all areas of the brain in both wild-type and 5xFAD mice (Fig. 10b), although overall densities were lower than the untreated mice in cortical regions (Fig. 10c, d; quantified in F). As demonstrated previously, untreated 5xFAD mice exhibit cortical plaques at 4 months of age, but 5xFAD mice devoid of microglia show reduced plaque formation (Fig. 4d, g, l). Furthermore, from our extended cohort of 5xFAD mice devoid of microglia, we know that plaque formation is severely diminished

in treated 5xFAD animals, even by 7 months of age (Fig. 4e, h, m). However, examination of microglia-repopulated 5xFAD brains revealed the appearance of robust plaque pathology (Fig. 10d) with plaque numbers being equal to the untreated 5xFAD mice (Fig. 10g), although average plaque volumes were smaller (Fig. 10h). Notably, vascular pathology was still present in the repopulated brains (Fig. 10d), showing that the reintroduction of microglia does not reverse the vascular deposition of Aβ, at least within this one-month timeframe. Repopulating microglia associate with the new plaques but do not appear to react to the vascular deposits (Fig. 10e). Moreover, GFAP+ astrocytes associated with plaques in both control and repopulated 5xFAD brains (Fig. 10i, j), but were absent in 5xFAD brains devoid of microglia. Thus, the reintroduction of microglia in the 5xFAD brain via CSF1R inhibitor withdrawal coincides with a full restoration of plaque pathology and implicate the reappearance of microglia in the brain with the seeding and formation of plaques.

**Discussion**

In this study, we sought to investigate the role of microglia throughout the onset and development of AD pathology in 5xFAD mice via the sustained depletion of microglia from the adult mouse brain for a period of ~6 months. While various methods of microglial ablation are available, the extent/duration of microglial depletion and the technical requirements necessary to achieve sustained microglial elimination prohibit the use of most depletion paradigms. For example, the $CX3CR1cre^{ER}xDTR^{ff}$ mouse model relies on administration of diphtheria toxin, which not only induces a cytokine storm, but also limits microglial elimination to 5 days[41,42]. The CD11b-HSVTK model requires intracerebroventricular infusion of ganciclovir in order to produce substantial microglial depletion, which in turn induces BBB damage and myelotoxicity, resulting in increased mortality following 4 weeks of ganciclovir treatment[11]. Additionally, clodronate liposomes can be administered to deplete microglia; however, this effect is short-lived and requires intrahippocampal infusion, as clodronate liposomes are incapable of crossing the BBB[43]. Previously, we discovered that microglia are critically dependent upon CSF1R signaling for their survival[17] and that CSF1R inhibitors serve as effective tools to achieve microglial depletion. In contrast to other methods, CSF1R inhibitor-induced microglial depletion is advantageous due to its (1) non-invasive route of administration, (2) independence from pharmacological drivers such as ganciclovir or tamoxifen, (3) lack of an inflammatory response (*i.e.*, cytokine storm) from surviving cells, (4) highly selective effects on the microglial compartment, (5) modifiable formulation, allowing for investigations into the contribution of peripheral myeloid populations on microglial responses, and (6) clinical utility, facilitating the translation of

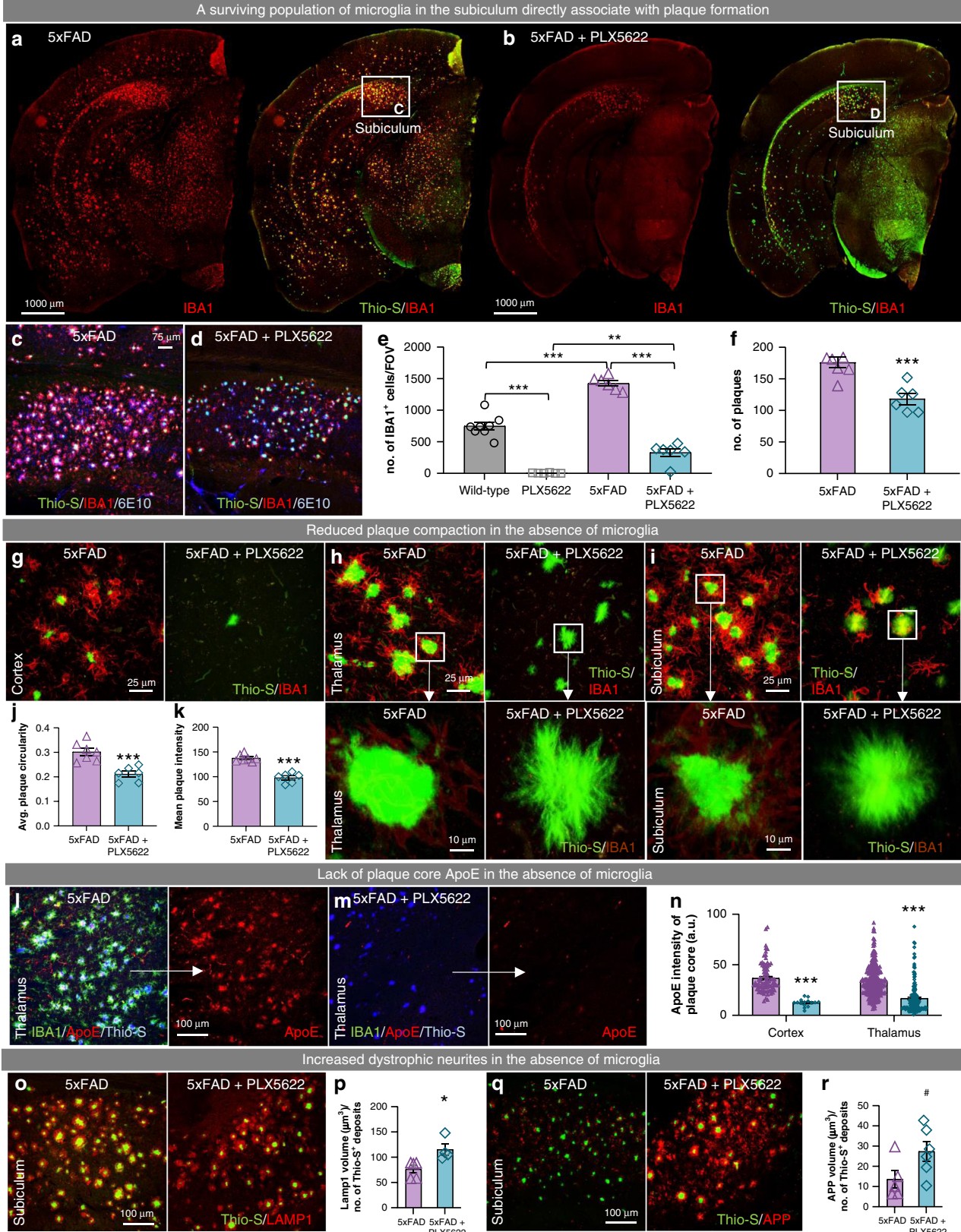

experimental findings. Most importantly, CSF1R inhibition is the only currently available method capable of achieving sustained long-term microglial elimination. Here, we sought to optimize a CSF1R inhibitor regimen to eliminate microglia brain-wide and maintain the absence of microglia for an unprecedented 6-month

treatment duration. To that end, we designed and created PLX5622, which shows high selectivity for the CSF1R (Supplementary Table 2), desirable PK characteristics across species (mouse, rat, dog, monkey; Supplementary Table 5) including oral bioavailability, a > 20% brain penetrance (Supplementary

**Fig. 6** Microglia facilitate plaque formation and compaction. **a**, **b**, Representative hemisphere stitches of 5xFAD and 5xFAD + PLX5622 mice stained for dense-core plaques (Thio-S in green) and immunolabeled for microglia (IBA1 in red). Scale bar = 1000 μm. **c**, **d** Confocal images of subicula stained with Thio-S for dense core plaques (green) and immunolabeled with IBA1 for microglia (red) and 6E10 for diffuse plaques (blue). Scale bar = 75 μm. **e** Quantification of IBA1$^+$ cells in the subiculum (WT v 5xFAD, $p < 0.001$; WT v PLX5622, $p < 0.001$; PLX5622 v. 5xfAD + PLX5622, p = 0.002; 5xFAD v 5xFAD + PLX5622, p < 0.001). Two-way ANOVA with Tukey's post hoc test; $n = 7$ for Wild-type, $n = 6$ for PLX5622, $n = 6$ for 5xFAD, $n = 6$ for 5xFAD + PLX5622. **f**, Plaque number within the subiculum is reduced by 33% in 5xFAD + PLX5622 mice compared to 5xFAD animals ($p < 0.001$). Two-tailed independent t-test; $n = 6$ for 5xFAD, $n = 6$ for 5xFAD + PLX5622. **g–i** Confocal images of dense-core plaques (Thio•S in green) and microglia (IBA1 in red) in the cortex, thalamus, and subiculum, respectively, of both 5xFAD groups showing an alteration in plaque morphology with CSF1R inhibitor treatment. Scale bar = 25 μm. Arrows point to zoomed images of dense-core plaques. Scale bar = 10 μm. **j**, **k**, Quantification of plaque circularity (**j**; 5xFAD v 5xFAD + PLX5622, $p < 0.001$) and Thio-S fluorescence intensity (**k**; 5xFAD v 5xFAD + PLX5622, $p < 0.001$). Two-tailed independent t-test; $n = 6$–7 for 5xFAD, $n = 6$ for 5xFAD + PLX5622. **l–m** Representative images from 7-month-old cohort immunolabled for ApoE (in red) and microglia (IBA1 in green) and stained for dense-core deposits (Thio-S in blue). Scale bar = 100 μm. **n** Quantification of ApoE immunoreactivity in plaque cores (cortex: $p < 0.001$; thalamus: $p < 0.001$). Two-tailed independent t-test; $n = 7$–10 for 5xFAD, $n = 7$ for 5xFAD + PLX5622. **o–r**, Immunolabeling and quantification of LAMP1 (**o**, **p**; $p = 0.020$) and APP (**q**, **r**; $p = 0.068$). Two-tailed independent t-test; $n = 5$ for 5xFAD, $n = 4$–5 for 5xFAD + PLX5622. Scale bar = 100 μm. Statistical significance is denoted by *$p < 0.05$, **$p < 0.01$, and ***$p < 0.001$. Statistical trends are denoted by #$p < 0.10$. Error bars indicate SEM

Table 6), and rapid/sustained elimination of microglia (Figs 2, 3). Using PLX5622, we were able to eliminate > 95% of microglia from the brains of 5xFAD mice prior to and during the formation of AD pathology – a treatment duration of up to 6 months - and elucidate their roles in plaque formation. Data presented here show that long-term PLX5622-mediated microglial depletion is highly robust, sustainable, and specific to the microglial compartment. These findings are demonstrated by extensive gene expression analyses, a lack of behavioral/cognitive impairments, and unaffected number of circulating immune cells. Together, these data demonstrate that PLX5622 is a useful compound for investigating microglial dynamics. Of note, loss of function mutations in the tyrosine kinase domain of the *Csf1r* are associated with the development of adult-onset leukoencephalopathy with axonal spheroids and pigmented glia[44], the onset of which is thought to derive from microglial phenotypic alterations[45]. However, we find here that long-term CSF1R inhibition (and consequential microglial loss) do not recapitulate these disease phenotypes, suggesting that the loss of a single CSF1R allele exerts different effects on the brain microenvironment relative to the loss of microglia.

The genetics of familial AD indicate that altered APP processing and plaque accumulation are critical for disease etiology[46]. In sporadic AD, plaques appear in the brain prior to any other overt pathologies[1] and precede cognitive deterioration and disease progression[47–49]. AD appears to be precipitated by the formation and development of plaques in the brain, and therefore, identifying the underlying biology of plaque formation is crucial to understanding and preventing disease onset. Genome-wide association studies have identified several genes associated with an increased risk for developing sporadic AD. Many of these risk variants are highly enriched in myeloid cells[3], indicating that myeloid biology appears to be a large contributor to the development of disease. Microglia are the resident myeloid cell of the CNS, and their activated presence surrounding plaques is a prominent feature of AD pathology. While microglia are phagocytes, in the later stages of disease, studies indicate that these cells do not clear Aβ or modulate plaque numbers/sizes in the brain[10–12]. Instead, evidence indicates that they play a role in synaptic damage and neuronal loss[10,49–51]. Examination of human brains (non-demented, high pathology non-demented, and Alzheimer's disease subjects), as well as mouse models of AD, reveal the accumulation of Aβ aggregates within the lysosomes of microglia unassociated with plaques. Thus, we hypothesized that neuronally-derived Aβ is internalized/aggregated within microglia, and that this material contributes to the initial formation of plaques. In a prior study we explored the roles of microglia in the later stages of AD, by eliminating microglia at a time point at

which 5xFAD mice had already developed extensive plaque pathologies. We found that the elimination of microglia consequently prevented the synaptic and neuronal loss, without affecting plaque loads[10], with subsequent studies confirming the contribution of microglia to neuronal loss[52,53]. In this study, we sought to identify whether microglia play a role in the initial deposition of plaques, via sustained treatment with PLX5622 prior to and during the period of plaque formation in the 5xFAD mouse model of AD. Here, we find that plaque formation is prevented in the absence of microglia, even over extended periods of time. While some plaques form in specific brain areas with treatment, we find that these few remaining plaques are associated with a small subset of microglia initially resistant to CSF1R-inhibition. These microglia are most apparent in the thalamus, retrosplenial cortex, and subiculum. Once formed, these plaques persist thereafter, even long after plaque-associated microglia are eliminated. Thus, our results demonstrate that plaque onset relies on the presence and activity of microglia. Additionally, as some microglia survive CSF1R inhibitor treatment, this approach also allows for the exploration of these remaining cells in the maintenance of plaque structure. Insights into the roles of microglia in plaque biology have been explored via the study of the myeloid cell-expressed gene *Trem2*, in which coding mutations thought to be loss of function are associated with increased risk of sporadic AD[54–56]. Recently, it was demonstrated that perturbation of TREM2 signaling impaired the association of microglia with plaques[36,37,57], revealing that plaque-associated microglia compact plaque cores and form a barrier that protects surrounding neurites from toxicity[35–37]. In line with these studies, we confirm and demonstrate that plaque compaction and barrier formation are natural functions of microglia in disease, rather than functions conferred by the loss of TREM2. In addition, we show that microglia are necessary for the development of plaques - these findings are supported by recent data showing that plaque seeding is accelerated in *Trem2* knockout mice[30], suggesting that microglial functions involved in the initial formation of plaques are related to TREM2 signaling. Collectively, these studies point to a role of microglia in plaque development, which is supported by our current findings.

By way of mechanism for plaque seeding and growth, microglia could be: (1) secreting factors that facilitate Aβ fibrillization[58], (2) physically forming plaque cores from extracellular Aβ via compaction[59], or (3) ingesting, aggregating, and modifying extracellular Aβ internally, which is eventually released to seed plaques, in line with in vitro data[60]. Indeed, low concentrations of Aβ can be taken up by cells and concentrated into acidic vesicles, yielding favorable conditions for the spontaneous aggregation of Aβ[15]. Moreover, excessive accumulation of Aβ within microglial

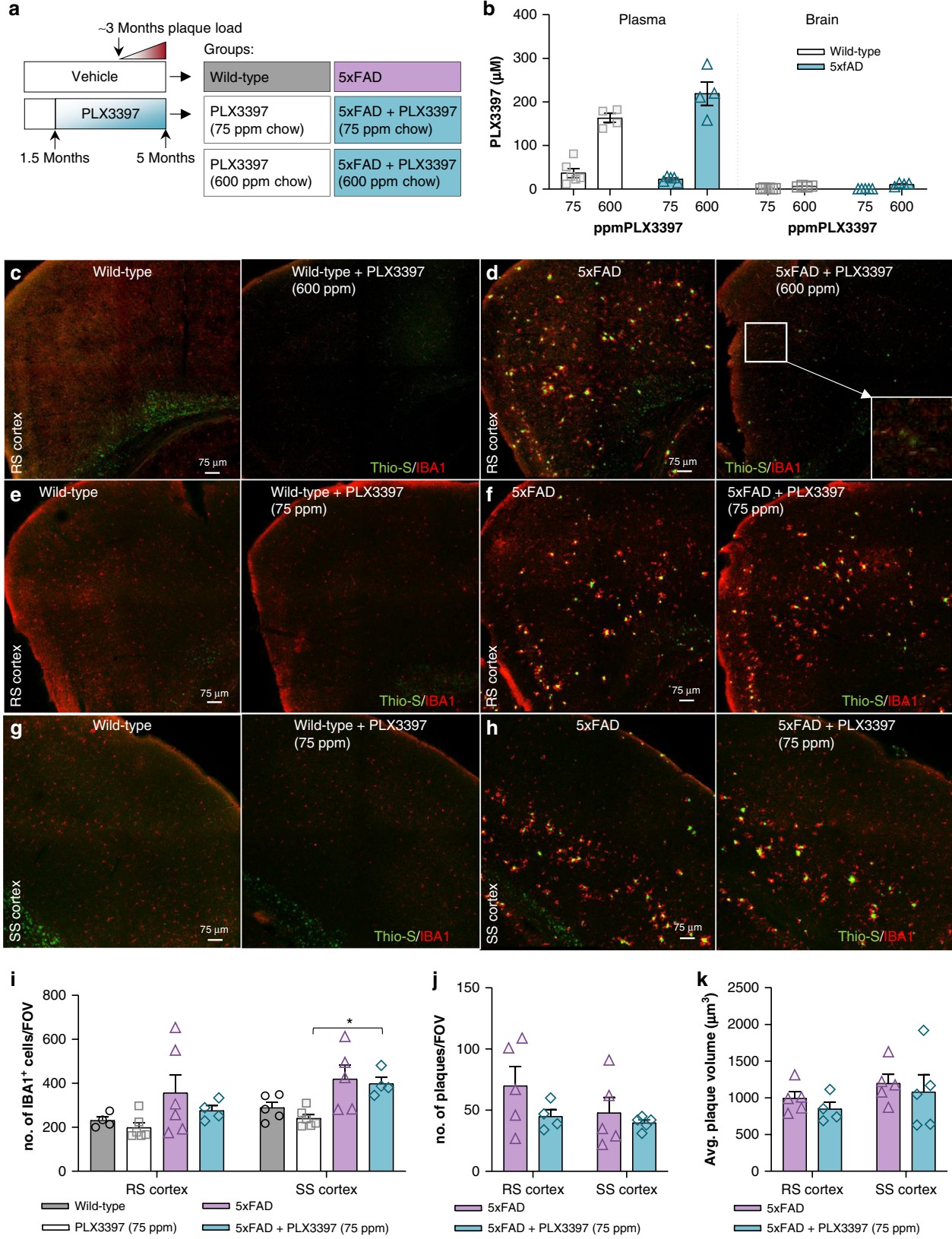

lysosomes can induce cellular death, potentially contributing to plaque expansion through the release of Aβ aggregates at the site of microglial death[59]. Here, we observe aggregated Aβ within the lysosome of ramified microglia distal to plaques in two different mouse models of AD (Fig. 1), as well as the human brain (Supplementary Figure 1). This observation is seemingly contrary to

the idea that the source of internalized microglial Aβ is plaque-derived. In accordance, we find that plaque growth ceases in the absence of microglia, consistent with recent data demonstrating microglial cellular death as a contributor to plaque growth[59], suggesting a mechanism by which microglia deliver Aβ to the plaque seed/core. These data could suggest that following the

**Fig. 7** Administration of an analogous CSF1R inhibitor, PLX3397 (75 ppm and 600 ppm), to 5xFAD mice. **a** Experimental design. **b** Terminal PK of wild-type and 5xFAD groups treated with PLX3397. **c, d** Confocal images of tissue stained for dense-core plaques (Thio-S in green) and immunolabeled for microglia (IBA1 in red) in 600 ppm PLX3397-treated and control mice. Scale bar = 75 μm. **e–h** Sections of the retrosplenial (RS) and somatosensory (SS) cortex, respectively, stained for dense-core plaques (Thio-S in green) and immunolabeled for microglia (IBA1 in red) in mice treated with control or 75 ppm PLX3397. Scale bar = 75 μm. **i** Quantification of IBA1$^+$ cell number in the RS and SS cortex. (SS Cortex: PLX5622 v 5xFAD + PLX5622, $p = 0.045$) Two-way ANOVA with Tukey's post hoc test; $n = 4$ for Wild-type, $n = 6$ for PLX3397, $n = 6$ for 5xFAD, $n = 4$ for 5xFAD + PLX3397. **j–k**, Quantification of cortical plaque number and volume, respectively, revealing no change in these measures with 75 ppm PLX3397 treatment in 5xFAD mice. Two-tailed independent $t$-test; $n = 4$–5 for 5xFAD, $n = 4$–5 for 5xFAD + PLX3397. Statistical significance is denoted by *$p < 0.05$. Statistical trends are denoted by $^\#p$ < 0.10. Error bars indicate SEM

uptake of freely available Aβ, microglia subsequently deposit aggregated Aβ into the extracellular space, thus contributing to the formation of a plaque. Therefore, microglia appear to have complex and contrasting roles within AD[61], namely their facilitation of plaque formation, plaque growth, plaque compaction, and neuronal damage.

To further delineate these roles, we performed gene expression analyses of wild-type and 5xFAD mice in cortex, hippocampus, and thalamus, in both microglia-intact animals and mice lacking microglia for the entirety of their adult lives. Notably, we find hippocampal gene expression is greatly influenced by the presence of AD pathology relative to the cortex and thalamus, despite abundant plaque load in all these regions. RNA-seq analysis revealed that nearly all significantly altered genes, in 5xFAD compared to WT mice, were associated with microgliosis in the cortex and thalamus, while the hippocampus displayed reductions in gene expression associated with neuronal and synaptic function. The absence of microglia prevented many of these changes in 5xFAD mice, despite the continued presence of hippocampal plaques, due to a small population of surviving myeloid cells in the subiculum. Although plaque-associated microglia may protect against local neurite damage, the presence of microglia is also required for the reduction in expression of synaptic and neuronal genes in the hippocampus associated with AD. Notably, the absence of microglia is associated with no alterations in immune- and synapse-related genes, suggesting that microglia mediate most AD pathology-induced changes in gene expression. Therefore, microglia appear to have detrimental and beneficial roles in a preclinical model of AD. Whether these effects are stratified into separate populations (*i.e.*, protective effects of plaque-associated microglia vs. harmful effects of non-plaque associated microglia, as recently suggested[39]) needs to be determined.

In conclusion, we have designed and created a specific CSF1R inhibitor, PLX5622, that allows for the sustained and specific elimination of microglia. This novel method of microglial depletion provided us with the means to eliminate microglia for the duration of AD pathogenesis. Ultimately, these data demonstrate that microglial elimination is associated with the prevention of plaque formation and the downregulation of hippocampal neuronal genes that occur in a preclinical model of AD progression. These results indicate that microglia appear to contribute to multiple facets of AD etiology – microglia appear crucial to the initial appearance and structure of plaques, and following plaque formation, promote a chronic inflammatory state modulating neuronal gene expression changes in response to Aβ/AD pathology.

## Methods

**Compounds**. PLX3397 (pexidartinib) was synthesized following the published procedure[62]. The synthesis of 2-fluoro-3-[(7-aza-5-methyl-3-indolyl)methyl]-6-[(5-fluoro-2-methoxy-3 pyridyl)methyl]aminopyridine (PLX5622), followed the procedure detailed in Supplementary Methods. For long term dosing, the compounds were formulated in AIN-76A standard chow by Research Diets Inc. at 1200 ppm (PLX5622), 300 ppm (PLX5622), 600 ppm (PLX3397), and 75 ppm (PLX3397). Instructions for the preparation of PLX5622 gavage dosing suspensions and chow are found in Supplementary Methods.

**Animal Treatments**. All animal experiments were approved by the UC Irvine Institutional Animal Care and Use Committee and were conducted in compliance with all relevant ethical regulations for animal testing and research. The 3xTg-AD[63] and 5xFAD[64] mouse models have been previously described in detail. For 3xTg-AD genotyping, the following primer sequences were used: APP Forward 5′ – GCT TGC ACC AGT TCT GGA TGG – 3′, APP Reverse 5′ – GAG GTA TTC AGT CAT GTG CT– 3′, PS1 Forward 5′ CAC ACG CAA CTC TGA CAT GCA CAG GC – 3′, and PS1 Reverse 5′ AGG CAG GAA GAT CAC GTG TTC AAG TAC – 3′. For 5xFAD genotyping, the primer sequences used were PS1 Forward 5′ - AAT AGA GAA CGG CAG GAG CA – 3′ and PS1 Reverse 5′ - GCC ATG AGG GCA CTA ATC AT – 3′. At the end of treatments, mice were euthanized via $CO_2$ inhalation and transcardially perfused with 1X phosphate buffered saline (PBS). For all studies, brains were removed, and hemispheres separated along the midline. Brain halves were either flash frozen for subsequent biochemical analysis or drop-fixed in 4% paraformaldehyde (PFA (Thermo Fisher Scientific, Waltham, MA)) for immunohistochemical analysis. Fixed half brains were sliced at 40 μm using a Leica SM2000R freezing microtome. The flash-frozen hemispheres were microdissected into cortical, hippocampal, and thalamic/striatal regions and then ground with a mortar and pestle to yield a fine powder. One-half of the powder from cortical and thalamic regions was homogenized in 500 or 250 μl Tissue Protein Extraction Reagent (TPER (Life Technologies, Grand Island, NY)), respectively, with protease (Roche, Indianapolis, IN) and phosphatase inhibitors (Sigma-Aldrich, St. Louis, MO) and centrifuged at 100,000 g for 1 h at 4 °C to generate TPER-soluble fractions. For formic acid-fractions, pellets from TPER-soluble fractions were homogenized in 500 or 250 μl 70% Formic Acid and centrifuged at 100,000 g for 1 h at 4 °C. Protein concentration in each fraction was determined via Bradford[65] and Lowry assays (Bio-Rad, per manufacturer's instructions). For RNA analyses, the second half of powder was processed with an RNA Plus Universal Mini Kit (Qiagen, Valencia, CA) according to the manufacturer's instructions.

**Human tissue**. Human postmortem tissue from non-demented, non-demented high pathology, and Alzheimer's disease subjects was obtained from the Alzheimer's Disease Research Center, UC Irvine with consent from the UC Irvine Ethics Committee. Neuropathological examination included Braak and Braak staging for plaques and tangles and diagnosis of neuropathological AD using National Institute on Aging-Reagan criteria. Human sections were obtained from 4% PFA drop-fixed regions of the middle frontal gyrus of the cortex and processed into 25 μm thick floating sections.

**Behavioral Testing**. ANY-Maze software was employed to video-record and track animal behavior. The following behavioral paradigms were carried out according to established protocols[10,17,64] and described briefly below:

Elevated plus maze (EPM): Mice were placed in the center of an elevated plus maze (arms 6.2 × 66 cm, with side walls 15 cm high on two closed arms, elevated 46 cm above the ground) for 5 min to assess anxiety.

Open field (OF): In brief, mice were placed in a white box (33.7 cm L×27.3 cm W×21.6 cm H) for 5 min to assess motor function and anxiety.

Spontaneous Alternation Y-Maze: Animals were placed in a Y-Maze (22 cm long×11.5 cm wide, sloping to 2.5 cm wide at the bottom with sidewalls 12.7 cm high) facing the back wall of the arm. Each mouse was allowed to explore the arena for 8 min. Visual extra- and intra-maze cues were provided to allow mice to create a spatial map of the maze. During the trial, the sequence of arm entries was scored live, as well as video-recorded for later analysis if needed. The number of triads was determined by counting the sequential entry into three different arms of the maze.

Morris water maze (MWM): Mice were placed in a plastic circular pool filled with opaque tap water (water mixed with water-soluble white paint). A white plastic platform was submerged 0.5 cm below the surface of the water. Distinct two-dimensional visual cues were positioned around the perimeter of the pool. The pool was visually divided into four quadrants, and the hidden platform was placed in one of these quadrants (#2), where it remained throughout the acquisition trials. Visual cues were taped to the walls to allow mice to create a spatial map during acquisition of the task. At the beginning of each acquisition trial, mice were placed on the platform for 10 s and then subsequently placed in the pool (quadrants were pseudorandomized) to swim freely for 60 s or until the platform was located. After all mice completed four trials, they were returned to their home cages. Twenty-four

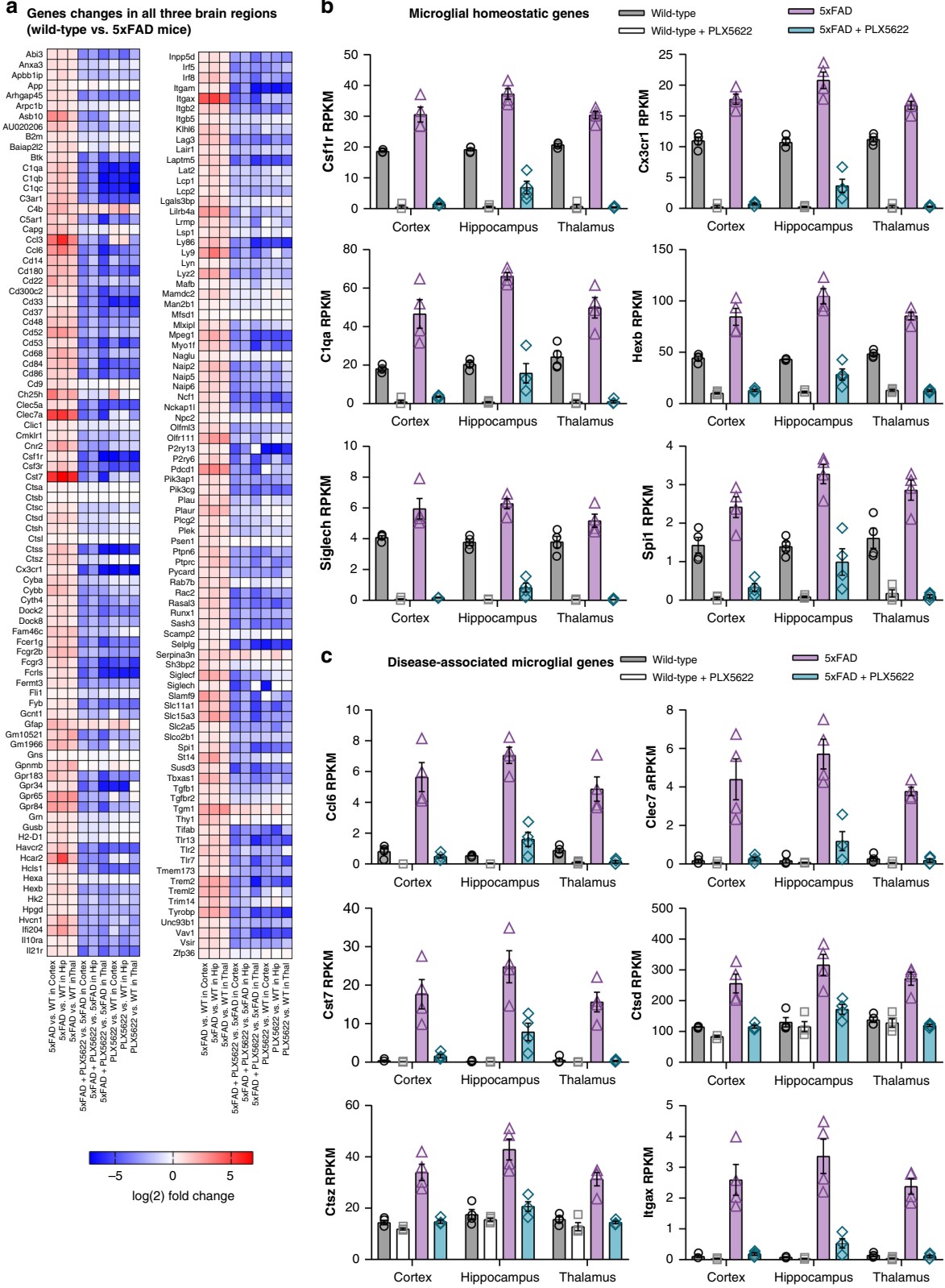

**Fig. 8** Remaining plaque-forming microglia in the microdissected hippocampus exhibit a DAM expression profile. **a** All gene expression changes where *p* < 0.05 for Wild-type (WT) vs. 5xFAD in all three brain regions shown as Log(2) fold change for each gene, for the 9 relevant comparisons (5xFAD vs. WT in cortex, hippocampus, or thalamus, 5xFAD + PLX5622 vs. 5xFAD in cortex, hippocampus, or thalamus, and WT + PLX5622 vs. WT in cortex, hippocampus, or thalamus). **b** RPKM values shown for a subset of the homeostatic microglial genes from (**a**), including *Csf1r*, *Cx3cr1*, *C1qa*, *Hexb*, *Siglech*, and *Spi1*. **c** RPKM values shown for a subset of the disease-associated microglial genes from (**a**), including *Ccl6*, *Clec7a*, *Cst7*, *Ctsd*, *Ctsz*, and *Itgax*. RPKM values for all genes/brain regions can be found at [https://rnaseq.mind.uci.edu/green/ad_plx/gene_search.php]. *n* = 4 per group for all analyses. Error bars indicate SEM

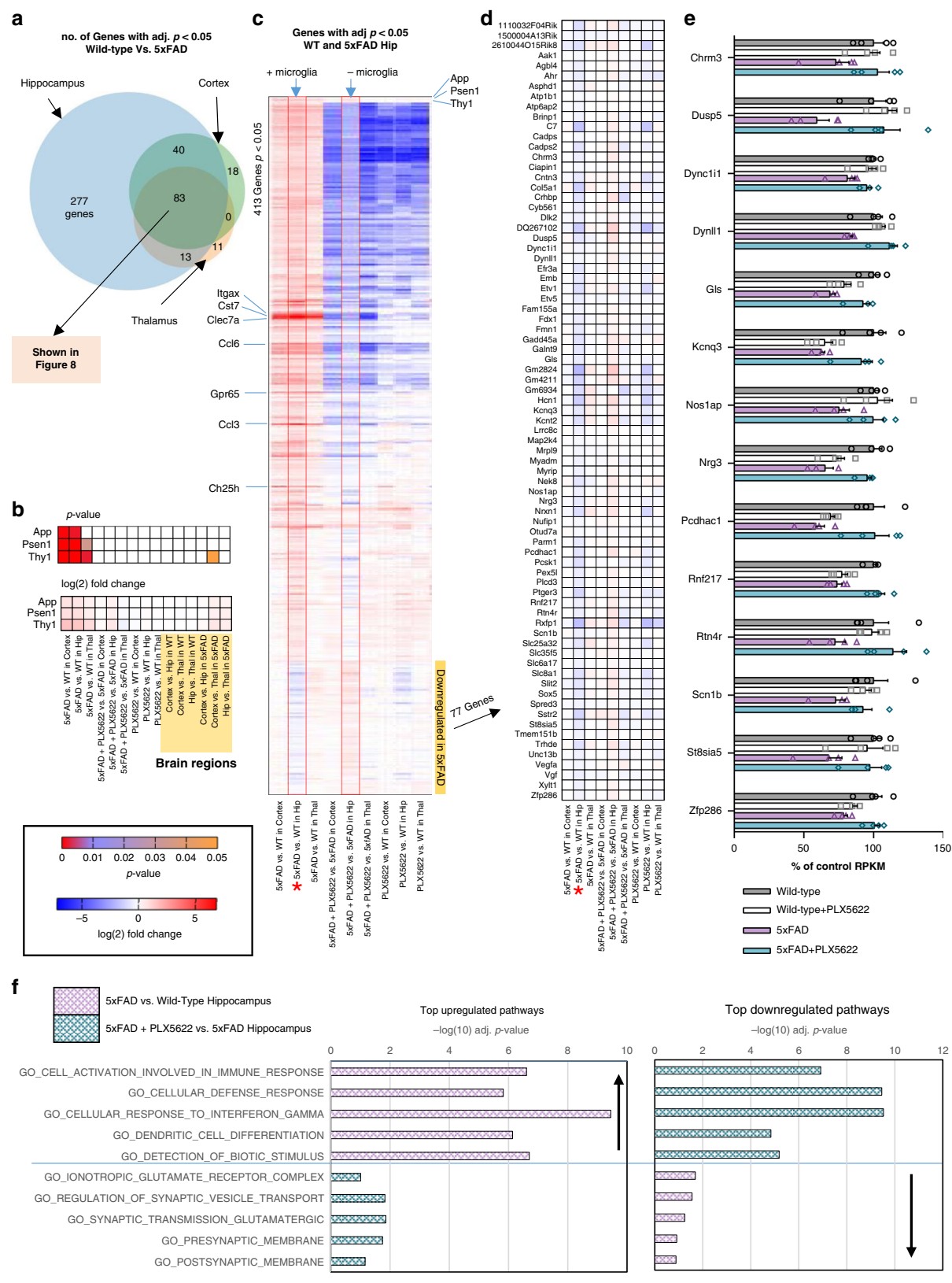

hours after the last day of acquisition testing, the platform was removed and the mice were subjected to a 60 s probe trial to assess spatial memory for the platform location.

Contextual Fear conditioning (CFC): Behavior was scored live and video-recorded using 1–0 sampling every 10 s, with a 1 denoting positive freezing behavior (total lack of body movement except respiration) and 0 indicating the absence of freezing behavior. In the training trial, mice were placed in a fear-conditioning chamber (Gemini, San Diego Instruments; 24.1 cm L×20.3 cm

W×20.3 cm H) to explore for 2 min before receiving one foot shock (3 s, 0.2 mA). Thirty seconds following the shock, animals returned to their home cage. Later 24 h, testing was conducted, whereby animals were placed in the chamber to explore for 5 min.

**Flow Cytometry.** Blood was drawn via cardiac puncture and incubated with ammonium-chloride-potassium lysing buffer for red blood cell lysis. Blood

**Fig. 9** Microglia mediate downregulation of neuronal/plasticity genes in the hippocampus in response to AD pathology. **a** Venn diagram showing the number of differentially expressed genes (adjusted (adj.) $p < 0.05$) for cortex, hippocampus, and thalamus for wild-type (WT) vs. 5xFAD mice. **b** Heatmap of the adj. p-value and log(2) fold change for all 9 comparison groups (5xFAD vs. WT in cortex, hippocampus, or thalamus, 5xFAD + PLX5622 vs. 5xFAD in cortex, hippocampus, or thalamus, and WT + PLX5622 vs. WT in cortex, hippocampus, or thalamus), as well as 3 comparisons between brain regions for both WT and 5xFAD mice, for the transgene components in 5xFAD mice (App, Psen1, Thy1). **c** Heatmap of all 413 gene expression differences identified in the hippocampus for wild-type vs. 5xFAD, expressed as log (2) fold change, and all 9 comparison groups included. Hippocampus for 5xFAD vs. WT and 5xFAD + PLX5622 vs. 5xFAD highlighted by red border, showing that most significant gene expression changes induced by pathology do not occur in the absence of microglia. **d** Downregulated genes in the hippocampus from (**a**), displayed as a heatmap of log(2) fold change differences for all 9 comparisons, showing that the same genes are not downregulated in the absence of microglia (5xFAD + PLX5622 vs. 5xFAD Hip). **e** RPKM values plotted on a log(2) scale for a subset of plasticity genes from (**d**). **f** Five significantly upregulated and downregulated pathways for 5xFAD vs. wild-type hippocampus (red), along with respective -log(10) p-values plotted: most upregulated pathways are related to immune function, while downregulated pathways are mainly associated with neuronal and synaptic activity. The same pathways are displayed for the comparison between 5xFAD + PLX5622 vs. 5xFAD hippocampus (yellow), showing that the absence of microglia prevents the upregulation of immune pathways and the downregulation of synaptic pathways. Expression difference denoted by *. Log(2) fold change and p-values indicated by respective scale bar. $n = 4$ per group for all analyses. Error bars indicate SEM

leukocytes were subsequently blocked with anti-mouse CD16/32 (14–0161–81; eBioscience) and stained with fluorophore-conjugated antibodies A700 anti-mouse CD45 (103127; BioLegend), PE anti-mouse CD11b (101208; BioLegend), APC-Cy7 anti-mouse CD11c (117323; BioLegend), PE-Cy7 anti-mouse CD4 (100421; Bio-Legend), APC anti-mouse CD8 (100711; BioLegend) and FITC anti-mouse CD19 (115505; BioLegend) as well as propidium iodide (PI; Invitrogen) for live/dead discrimination. Data were acquired on a BD FACSAria II and analyzed using the FlowJo software to assess peripheral cell counts. Gating was performed following routinely used protocols, using FSC/SSC to exclude dead cells, cell debris and doublets and propidium iodide to identify dead cells. At least 250,000 events were captured by the cytometer and 30,000 single cells analyzed per group. Cells were gated on CD45$^+$PI$^-$ live singlets. Dead cells were excluded as propidium iodide (PI)-positive. Populations of live singlets were then gated was percentages of CD45 + PI- cells.

**Confocal Microscopy.** For Thioflavin-S (Thio-S) staining, free-floating sections were washed with 1X PBS (1 × 5 min), followed by dehydration in a graded series of ethanol (100%, 95%, 70%, 50%; 1 × 3 min each). The sections were incubated in 0.5% Thio-S (in 50% ethanol, Sigma-Aldrich) for 10 min. This was followed by 3 × 5 min washes with 50% ethyl alcohol and a final wash in 1X PBS (1 × 10 min). For 6E10 and Aβ$_{42}$ immunohistochemistry, sections were briefly rinsed in 1X PBS (1 × 5 min) and underwent antigen-retrieval via incubation in 90% Formic Acid (1 × 4 min), followed by 4 × 4 min washes in 1X PBS. Following Thio-S staining or formic acid pretreatment (if required), sections underwent a standard indirect immunohistochemical protocol. To that end, free-floating sections were washed with 1X PBS (1 × 5 min), and immersed in normal serum blocking solution (5% normal serum with 0.2% Triton-X100 in 1X PBS) for 60 min. Primary antibodies and dilutions used are as follows: anti-ionized calcium-binding adapter molecule 1 (IBA1; 1:1000; 019–19741; Wako, Osaka, Japan), anti-Aβ$_{1-16}$ (6E10; 1:1000; 803001; BioLegend, San Diego, CA), anti-A11 oligomers (A11; 1:100; AHB0052; Thermo-Fisher Scientific), anti-amyloid fibrils OC (OC; 1:100; AB2286; EMD Millipore; Burlington, MA), anti-Aβ$_{1-42}$ (1:200; ab10148; Abcam), anti-CD68 (1:500; MCA1957; Bio-Rad, Hercules, CA), anti-CD11b (1:50, MCA711; Bio-Rad), anti-β−amyloid [pyroglutamate-3] (p3GluAβ; 1:500; NBP1–44048; Novus Biologicals, Littleton, CO), anti-lysosomal associated membrane protein 1 (LAMP1; 1:200; sc-20011; Santa Cruz Biotechnology; Dallas, TX), anti-amyloid precursor protein, c-terminal (APP; 1:500; 171610; Calbiochem, San Diego, CA), anti-S100β (1:200; ab52642; Abcam, Cambridge, MA), anti-apolipoprotein E (ApoE; 1:100; ab1906; Abcam), anti-claudin-5 (1:500; 35–2500; Invitrogen) and anti-glial fibrillary protein (GFAP; 1:1000; ab4674; Abcam). Thioflavin-S (Thio-S; Sigma-Aldrich) staining was carried out as described above. Amylo-Glo (TR-300-AG; Biosensis, Thebarton, South Australia, AU) staining was performed according to the manufacturer's instructions. Prussian blue staining (ab150674; Abcam) was carried out according to manufacturer's instructions. Total microglia and plaque counts/volumes were obtained by imaging comparable sections of tissue from each animal at the ×10, ×20, or ×63 objective, at multiple z-planes, followed by automated analyses using Bitplane Imaris 7.5 spots or surfaces modules, respectively. Plaque circularity was evaluated using ImageJ software. ApoE intensity within plaque cores was measured using Bitplane Imaris 7.5 surfaces module. Hemisphere stitches were imaged using StereoInvestigator Software on a Zeiss Imager.M2 Stereology Scope. Three-dimensional reconstruction of microglia and plaques was generated using the surfaces module of Bitplane Imaris 7.5.

**Immunoblotting.** Equal amounts of TPER-soluble protein (20 μg) were separated by sodium dodecyl sulfate–polyacrylamide gel electrophoresis (SDS□PAGE) on a 4–12% Bis/Tris gel (Bio-Rad, Hercules, CA), transferred to 0.20 μm nitrocellulose membranes, blocked for 1 h in 5% (vol/vol) nonfat milk or bovine serum albumin (BSA) in Tris□buffered saline (pH 7.5) supplemented with 0.2% Tween20. Antibodies and dilutions used include anti-6E10 (6E10; 1:1000; 803001; BioLegend), anti-APP C-Terminal (APP; 1:1000; 171610; Calbiochem, San Diego, CA) for C99

and C83, anti-ADAM10 (1:1000; ab1997; Abcam), anti-β-secretase 1 (BACE1; 1:1000; ab2077; Abcam), anti-Presenilin-1 (PS1; 1:1000; ab76083; Abcam), anti-Presenilin enhancer 2 (PEN2; 1:500; ab18189; Abcam), anti-A11 (A11; 1:1000; AHB0052; Thermo-Fisher Scientific), and anti-β-actin (1:10,000; MA5–15739; Sigma-Aldrich). Blots were developed using Pico/Dura Western Blotting Detection System (Pierce) and exposed to film for images. Quantitative densitometric analyses were performed on digitized images of immunoblots using the NIH program ImageJ and band densities were normalized to β-actin. Uncropped scans of the blots are presented in Supplementary Figure 10A and 10B and provided in Source Data file.

**Aβ ELISA.** Isolated protein samples were transferred to a blocked MSD Human (6E10) Aβ triplex ELISA plate (Aβ$_{1-38}$, Aβ$_{1-40}$, Aβ$_{1-42}$) and incubated for two hours at room temperature with an orbital shaker. The plate was then washed and measurements obtained using a SECTOR Imager 2400, per the manufacturer's instructions (Meso Scale Discovery, Gaithersburg, MD).

**RNA Sequencing.** Whole transcriptome RNA sequencing (RNA-Seq) libraries were produced from Wild-type (WT), PLX5622, 5xFAD, and 5xFAD + PLX5622 mice treated from 1.5 to 7 months of age, brains that were microdissected into Cortex, Hippocampus, and Thalamus ($n = 4$/group for each of the 3 brain regions = 48 total samples). Briefly, 100–600 ng of RNA were depleted of ribosomal RNA, fragmented, reverse transcribed and ligated to indexed sequencing adapters using the KAPA RNA HyperPrep Kit with RiboErase. Amplified libraries were combined into 4 pools of 12 libraries and sequenced on 4 lanes of a HiSeq4000 producing 50 bp single-end reads. This work used the Vincent J. Coates Genomics Sequencing Laboratory at UC Berkeley, supported by NIH S10 OD018174 Instrumentation Grant.

Reads were mapped to the reference mouse genome (mm10) using STAR[66] aligner and quantified with the featureCounts function of the Rsubread[67] package in R[68]. After filtering out low-count genes, count distributions were scaled using the calcNormFactors function of the edgeR[69] package. Transgene/human alignments were not filtered out from mouse reads. Principal component analysis was performed using the plotMDS function of the limma[70] package in R. Normalized counts were prepared and fitted to linear models using the voom and lmfit functions of limma respectively. Gene ontology gene sets for mouse were downloaded from http://bioinf.wehi.edu.au/software/MSigDB/. Competitive gene set testing was performed using the camera function (limma package), which calculates two-tailed p-values and FDR adjusted $p$-values ($p < 0.1$) using the Benjamini and Hochberg method.

**Nanostring analysis.** RNA was extracted and purified from frozen half brains using an RNA Plus Universal Mini Kit (Cat. #73404, Qiagen). For nCounter© analysis, total RNA was diluted to 20 ng/μl and probed using a mouse nCounter© Neuropathology Panel (Nanostring Technologies, Seattle, WA, USA). Counts for target genes were normalized to the best fitting house-keeping genes as determined by nSolver software.

**Statistics.** Every reported $n$ is the number of biologically independent replicates. No statistical methods were used to predetermine sample sizes; however, our sample sizes are similar to those reported in recently published similar studies[10,28]. Behavioral, biochemical, and immunohistological data were analyzed using either two-tailed independent-samples $t$-test (Control vs. Wild-type (WT) or 5xFAD vs. 5xFAD + PLX3397/PLX5622) in Microsoft Excel or as a one- (Diet: Control vs. PLX3397/PLX5622 vs. Repopulation) or two-way ANOVA (Diet: Control vs. PLX3397/PLX5622 and Genotype: WT vs. 5xFAD) using GraphPad Prism Version 6 (La Jolla, CA). Tukey's post hoc tests were employed to examine biologically relevant interactions from the two-way ANOVA regardless of statistical significance of the interaction. For RNA analyses, moderated $t$-statistics and

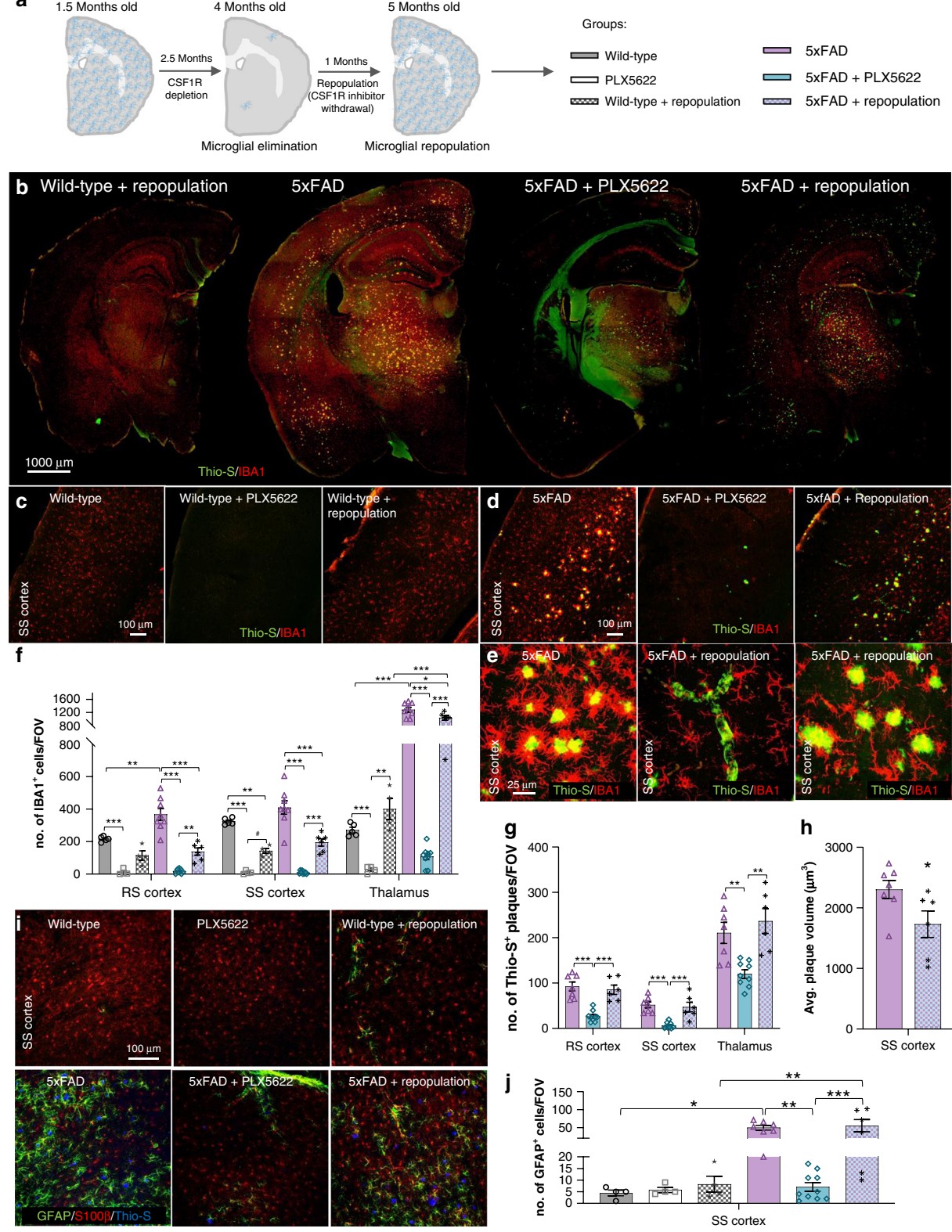

corresponding p-values (raw) were calculated for each of the relevant comparisons for each gene (5xFAD vs. WT in cortex, hippocampus, or thalamus, 5xFAD + PLX5622 vs. 5xFAD in cortex, hippocampus, or thalamus, and WT + PLX5622 vs. WT in cortex, hippocampus, or thalamus), as well as the comparisons between brain regions (cortex vs. hippocampus, cortex vs. thalamus, and hippocampus vs. thalamus for both WT and 5xFAD mice) using the eBayes function in limma. Raw p-values were adjusted across all comparisons to account for multiple testing using the decideTests function (Benjamini-Hochberg method). Genes with an adjusted p-value below 0.05 were considered differentially expressed. Data distribution was

assumed to be normal, but this was not formally tested. Symbols denote significant differences between groups: $*p < 0.05$, $**p < 0.01$, and $***p < 0.001$. Statistical trends are accepted at $p < 0.10$ ($^{\#}$). Data are presented as raw means and standard error of the mean (SEM).

**Reporting summary**. Further information on research design is available in the Nature Research Reporting Summary linked to this article.

**Fig. 10** Microglia seed plaques. All analyses listed in respective order for retrosplenial (RS) cortex, somatosensory (SS) cortex, and thalamus.
**a** Experimental design: 1.5-month-old wild-type (WT) and 5xFAD mice were administered PLX5622 (1200 ppm in chow) until 4 months of age. Diet was withdrawn for 1 month to allow microglial repopulation. This figure was created with images adapted from Servier Medical Arts and is licensed under the Creative Commons Attribution 3.0 Unported License [https://creativecommons.org/licenses/by/3.0/]. **b** Representative hemisphere stitches of sections stained for dense core plaques (Thio-S in green) and immunolabeled for microglia (IBA1 in red). Scale bar = 1000 μm. **c–e** Images of dense-core plaque staining (Thio-S in green) and microglia immunolabeling (IBA1 in red). Scale bar = 100 μm for C,D; 25 μm for E. **f**, IBA1$^+$ cell number (WT v 5xFAD, $p$ = 0.001, NS, $p < 0.001$; WT v PLX5622, $p < 0.001$, $p < 0.001$, $p < 0.001$; 5xFAD v 5xFAD + PLX5622, $p < 0.001$, $p < 0.001$, $p < 0.001$; PLX5622 v Repopulation, NS, $p = 0.054$, $p = 0.008$; 5xFAD + PLX5622 v 5xFAD + Repopulation, $p = 0.006$, $p < 0.001$, $p < 0.001$; WT v Repopulation, NS, $p = 0.002$, NS; 5xFAD v 5xFAD + Repopulation, $p < 0.001$, $p < 0.001$, $p = 0.027$; Repopulation v. 5xFAD + Repopulation, NS, NS, $p < 0.001$). Two-way ANOVA with Tukey's post hoc test; $n = 5$ for Wild-type, $n = 4$ for PLX5622, $n = 4$ for Wild-type + Repopulation, $n = 8$ for 5xFAD, $n = 9$ for 5xFAD + PLX5622, $n = 6$ for 5xFAD + Repopulation. **g** Quantification of Thio-S$^+$ plaque number (5xFAD v 5xFAD + PLX5622, $p < 0.001$, $p < 0.001$, $p = 0.007$; 5xFAD + PLX5622 v 5xFAD + Repopulation, $p < 0.001$, $p < 0.001$, $p = 0.001$; 5xFAD v 5xFAD + Repopulation, NS, NS, NS). One-way ANOVA with Tukey's post hoc test; $n = 7$ for 5xFAD, $n = 9$ for 5xFAD + PLX5622, $n = 6$ for 5xFAD + Repopulation. **h** Mean cortical plaque volumes (5xFAD v 5xFAD + Repopulation, $p = 0.048$). Two-tailed independent $t$-test; $n = 7$ for 5xFAD, $n = 6$ for 5xFAD + Repopulation. **I** Images of dense-core plaques (Thio-S in blue) and astrocytes with GFAP (in green) and S100β (in red). Scale bar = 100 μm. **j** GFAP$^+$ astrocyte number in the SS cortex (WT v 5xFAD, $p = 0.010$; 5xFAD v 5xFAD + PLX5622, $p = 0.002$; 5xFAD + PLX5622 v 5xFAD + Repopulation, $p < 0.001$; WT + Repopulation v 5xFAD + Repopulation, $p < 0.009$). Two-way ANOVA with Tukey's post hoc test; $n = 4$ for Wild-type, $n = 4$ for PLX5622, $n = 4$ for Wild-type + Repopulation, $n = 7$ for 5xFAD, $n = 10$ for 5xFAD + PLX5622, $n = 6$ for 5xFAD + Repopulation. Statistical significance is denoted by *$p < 0.05$, **$p < 0.01$, and ***$p < 0.001$. NS, not significant ($p > 0.10$). Error bars indicate SEM

## Data availability

A searchable database with all RPKM values can be found at [http://rnaseq.mind.uci.edu/green/ad_plx/gene_search.php]. The atomic coordinates and structure factors of CSF1R in complex with PLX5622 (PDB accession code: 6N33) can be downloaded from the website of Protein Data Bank, [www.rcsb.org]. The RNA-Seq data has been deposited in GEO (accession number: GSE134151). Uncropped blots can be found in Supplementary Figure 10 and the Source Data file. Additional data that support the findings of this study are available from the corresponding author upon reasonable request.

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

## Acknowledgements

This work was supported by the National Institutes of Health under awards R01NS083801 (NINDS), R01AG056768 (NIA), and P50AG016573 (NIA) to K.N.G., AARF-16–442762 (Alzheimer's Association) to L.A.H. and F31AG059367 (NIA) and T32AG00096 (NIA) to E.S. The content is solely the responsibility of the authors and does not necessarily represent the official views of the National Institutes of Health. X-ray diffraction data were collected at the Advanced Light Source, which is a DOE Office of Science User Facility under contract no. DE-AC02–05CH11231. We thank Dan Hoang, Karla Abad-Torrez, Ken Nguyen, and Dr. Michael Phelan for the setting up of gene expression data.

## Author contributions

E.S. conceived and performed experiments, analyzed data, and wrote the manuscript. P.L. Severson provided expertize and analyzed data. L.A.H. and J.C. performed experiments and analyzed data. J.Z., E.A.B., Y.Z., W.S., and J.L. designed and performed experiments and analyzed data. N.Y.P. performed analyzed data. G.H., A.R., G.T., J.W., M.N., P.I., C.Z., and G.B. designed and performed experiments and analyzed data. P.Singh, S.B. and B.L.W. provided expertize and feedback. K.N.G. conceived experiments, wrote the manuscript, provided supervision, and secured funding.

## Additional information

**Competing interests:** P.L.Severson, J.Z., E.A.B., Y.Z., W.S., J.L., G.H., A.R., G.T., J.W., M.N., P.Singh, S.B., P.I., C.Z., G.B., and B.L.W. are employees of Plexxikon Inc.; the remaining authors declare no competing interests.

