## [Peer Review File · Nature Communications]

Editorial Note: Parts of this Peer Review File have been redacted as indicated to maintain confidentiality.

Reviewers' comments:

Reviewer #1 (Remarks to the Author):

In this study, Spangenberg et al. designed a new CSF1R-selective inhibitor (PLX5622) that rapidly ablates the vast majority of microglia in the mouse brain. The authors then used PLX5622 to eliminate microglia for an extended period of time in a mouse model of Alzheimer's disease (5xfAD) and monitored the effect on Alzheimer's disease pathology and gene expression. They found that eliminating microglia largely prevented β -amyloid plaque formation in the parenchyma. Instead, in the absence of microglia, β -amyloid was found to be deposited along the cortical blood vessels. Furthermore, microglia ablation in the 5xfAD model prevented the upregulation of inflammatory genes and a downregulation of synaptic genes usually observed in 5xfAD animals in the presence of microglia.

This is an informative study that illustrates how microglia influence multiple aspects of Alzheimer's disease pathology, most notably demonstrating that microglia facilitate plaque formation, growth and compaction. However, there are a number of concerns that need to be addressed, before the manuscript can be considered for publication.

Major points:

It has been reported previously that the acute ablation of microglia triggers ataxia-like behavior. "Microglia-depleted mice progressively developed severe motor coordination dysfunction similar to spinocerebellar ataxia, including kyphosis, loss of balance and hind limb clasping." (Rubino, ..., Butovsky, Lassmann & Weiner Nature Communications volume 9, Article number: 4578 (2018)). These phenotypes would likely influence the behavioral assays the authors performed in this study. Therefore, the authors should test whether depletion of microglia with PLX5622 causes any of the phenotypes described above and determine whether their ablation system also affects locomotion as measured by rotarod testing.

- Line 280. The authors have to comment on the why there is such a huge difference in green signal (IBA1 staining) in the Wildtype+PLX5622 versus 5xfAD+PLX5622. We expect that these two conditions would have almost no green signal, especially from their statistical analysis, but instead the Wildtype looks+PLX5622 looks high in green signal (Figure 3C). As the manuscript stands, the small inset appears to be selected with the bias towards of giving the lowest possible green signal value in Wildtype+PLX5622.

- Line 337: "The absence of microglia did not significantly alter the levels of A β 38, A β 40, or A β 42 in detergent-soluble or -insoluble cortical and thalamic homogenates in either the 4- or 7- month cohorts (Fig. 5A-H)." This is not an accurate statement. There is a significant reduction in 5B. In Fig 5A and C there are trends in this direction as well. When looking at the individual data points, the lack of difference is likely a matter of too few data points. In some cases, only 3 mice are used (Fig 5D) and this will bias the data to lack of significant difference. In all figures of the paper pertaining to histology and other kinds of protein quantification, the authors should use at least 7 mice (or data points) in order to perform a meaningful statistical analysis. This is also obvious in figure 5L where signal intensity of BACE1 or PS-1 could become significant, if more than 4 data points were collected. Looking at the blot in figure 5K, C-term PS1 looks to be increased by PLX5622 treatment in Wildtype mice, but a statistical analysis with such few data points will not reflect this.

Minor points:

- Line 141, the authors should specify the kind of fixation used (drop-fixed as in line 91, or intracardial PFA?).

- Line 233. The authors describe the accumulation of Thio-S+ material in non-plaque associated microglia. One drawback about thioflavin S staining is the lack of specificity as it non-selectively binds beta sheet contents of proteins. The authors should confirm that non-plaque associated microglia indeed contain intracellular β -amyloid aggregates using an antibody detecting β -amyloid.
- Line 292. The authors claim that long-term absence of microglia is not detrimental to murine cognitive function. It is possible that the performance in the MWM test is not affected but other tests would reveal a difference. The authors may consider to qualify this statement or to perform additional behavioral tests.
- Line 342: "Investigations into... found no significant changes... (Fig 5I,J)." From the figure 5I it looks like the A11 intensity is much weaker in 5XFAD+PLX5622 than in untreated 5XFAD mice. Perhaps a more representative image would be preferable.
- Line 378" ... and lacked the homogenous staining intensity of plaques in microglia-intact 5xfAD mice." The authors need to clarify exactly how staining intensity was quantified in Fig 6I. From the micrographs, the green signal intensity looks virtually identical, at least in the center or the plaques (Figure 6F1 and 6G1).
- Line 395 "Treated 1.5-month-old 5xfAD mice with this formulation until 5 months of age" versus line 398 "Treating 1.5-month-old 5xfAD mice for 5 months". This is probably a typo, but the authors should clarify if the mice are 5 months old or treated for 5 months with these drugs.
- Figure 3D-K. The authors should use consistent data display design, rather than combining bars and violin plots. The chosen colors (gray and purple shades) of the different groups are also not easily discernible. The data would be clearer with colors like black, white, red, blue, green and so on. There is also no reason to use different symbols for the different bars since these are clearly separated by distance. In order to visualize the individual data points in a clear manner, the authors should consider using open circles in all cases with a thin border that clearly contrasts in color to the bars.
- The legends of the figures describe n's used as a range, e.g. line 908 "n=8-12 for Wildtype" etc. Exact numbers should be given in all cases.
- Line 408: "Gene expression analyses of Wild-type and 5xfAD mice in the presence and absence of microglia, across brain regions" The authors should clarify if the mice were ablated with PLX3397 or PLX5622.
- Line 416. "extracted,and..." space is missing
- Line 575: "... small subset of surviving CSF1R-resistant microglia". Perhaps a better wording would be "... small subset of surviving microglia resistant to CSF1R-inhibition"
- Fig10J: Because there are such large differences between wildtype and 5xfAD in "# of GFAP+ Cells / FOV", the authors should consider splitting the graph, so that the reader can better see the data points of the wildtype mice in the three conditions. The authors should also comment on the increase in number of GFAP positive cell sin the repopulating wildtype mice. Is this something observed in other brain regions? The same analysis should be performed for "RS cortex" and "Thalamus" for consistency with previous figures.

Reviewer #2 (Remarks to the Author):

Thank you for the opportunity to review Nature Communications Manuscript#: NCOMMS-19-01388-T.

Overall the manuscript is well-written and presents interesting scientific hypotheses that are backed by data. The authors demonstrate the complete, long-term elimination of microglia in WT and 5xfAD mice, which is accomplished by chow administration of selective and brain penetrant CSF1R inhibitor PLX5622. This level and length of microglial depletion is unprecedented, and enables the detailed evaluation of changes to AD-associated pathology, including brain plaques and gene expression. This data is used to generate hypotheses about the potential causative role of microglia in the progression of AD pathology based on this mouse model data. The results are of

immediate interest for the neuroscience community, and the quality of data is excellent.

There are a few conclusions needing attention.

(1) In general, although the association of microglial depletion in the brain with findings is plausible, in many instances the absence of microglia is not proven to be definitively causative of observed effects. The manuscript should be re-worded to that effect. Also the findings are in a mouse model, not in AD itself and that should be clarified. Some examples with suggested updates are given:

- Line 310: "suggest that microglia may be critical regulators"
- Line 382: "some threshold density of microglial cells may be required"
- Line 386: "these results are consistent with prior studies"
- Line 446: "Microglial depletion results in the downregulation of"
- Line 465: reword as "in the absence of microglia, AD-pathology induced down regulation of neuron-associated genes was prevented"
- Line 467: reword as "microglia in the 5xfAD brain may drive downregulation"
- Line 515: remove word "directly"
- Line 552: ideal is overly strong - suggest "a useful tool compound for investigating"
- Line 561: "myeloid biology appears to be a large contributor"
- Line 610: "our study suggests that the presence"
- Line 627: "the absence of microglia is associated with no alterations"
- Line 628: replace demonstrating with suggesting
- Line 629: "and beneficial roles in a preclinical model of AD"
- Line 636: "microglial elimination is associated with prevention of formation of plaques"
- Line 637: "that occur in a preclinical model of AD progression"
- Line 638: "microglia appear to contribute to the multiple facets"

(2) There are instances where figure information and/or statistics do not agree with what is stated in the manuscript text.

- Line 320: "across all brain regions (Fig. 4N)" – figure shows NS in the cortex
- Line 351: "no overt changes" (Fig 5M) – however significant changes observed in *Inpp5d*, *Spi1*, *Trem2*, not mentioned in text
- Lines 437-442: only 3 genes (*Ccl6*, *Clec7a*, *Cst7*) appear to be present in the hippocampus but not cortex or thalamus in the PLX5622-treated 5xfAD animals – "these genes" implies that all 6 genes are present, as well as the other ~14 genes listed in the text. Please clarify wording.
- Line 449, 453, 462 – I count 413 genes (not 414) in Fig 9A
- Figure 10E – this appears to be 5xfAD + PLX5622 but is labeled as 5xfAD + Repopulation

(3) The structural explanation for the observed improved broad kinome selectivity of PLX5622 is unpersuasive, and no rationale for the improved brain penetration of PLX5622 is provided.

- Lines 252-255 and 861-863: A fluorine atom has approximately the same atomic radius as a hydrogen atom. The steric argument that a bulkier cysteine in KIT and FLT3 is the rationale for greater selectivity does not make sense.
- Lines 255-259 and 865-866: is there x-ray data to support the stabilization of the allosteric pocket of CSF1R? If not, then need to specify that this is a hypothesis.
- Figure 2B shows an H-bond interaction of the pyridine N with the protein backbone but this is not mentioned in the text or caption. The key hinge-binding interactions are also not described.
- Lines 260-262: what physicochemical characteristics of PLX5622 are responsible for the increased brain penetration? Was this predicted?

(4) It is unclear how testing the less selective (and structurally highly similar) reagent PLX3397 rules out "potential confounding off-target effects." (Lines 392-393).

- Considering the cellular activity (THP1 = 0.032 μ M), why at a concentration of ~1 μ M in the brain is no effect on microglia observed? What is the unbound brain concentration & is there high brain protein binding?
- Line 596 – what do you mean by "off target CSF1R inhibition"? Suggest striking this.

(5) Parts of the discussion section are confusing.

- Lines 578 -586 - what data suggests an interplay between ApoE and microglia facilitating plaque

pathogenesis, other than a correlation in the timing?

- Lines 602: include a cross reference for the 2nd model of AD (Fig 1, A-F), and for the human brain (Fig 1, K-O).

Previous literature is referenced appropriately, with two suggestions.

1. A recent review of AD advancements with regard to microglia would be appropriate to include, e.g. Brit. J. Pharm. 2018, "The involvement of microglia in Alzheimer's disease: a new dog in the fight." DOI:10.1111/bph.14546
2. Description of the phenotype of CSF1R-deficient mice and humans with CSF1R mutations would enhance the manuscript.
 - a. E.g. (mice) PLoS One 2011; "Absence of colony stimulation factor-1 receptor results in loss of microglia, disrupted brain development and olfactory deficits." DOI: 10.1371/journal.pone.0026317.
 - b. There are multiple examples of CSF1R mutations in humans that results in hereditary diffuse leukoencephalopathy with spheroids; this example focuses on impact on microglia. Annals of Neurology 2016. "Characteristic microglial features in patients with hereditary diffuse leukoencephalopathy with spheroids." DOI:10.1002/ana.24754

A few additional recommendations, for clarity:

Line 32 – mice (not animals)

Line 45 – in the AD brain of rodents (otherwise may imply human, and minimal human data is shown)

Line 74 – of multiple facets of AD pathology (since this is a model, and not the actual disease)

Line 97/98 – are the specific protease and phosphatase inhibitors obvious to those skilled in the art?

Lines 295/296, also line 588 – no evidence of "shuttling" of A β is provided, simply the finding that A β is present in the vasculature. Could there be sequestering of A β ? Or some other interpretation? This should be re-worded for accuracy. The discussion in lines 605-608 are reasonable.

Lines 332, 336 – "Data not shown." Minimally should include this data in supporting information.

Line 397 – Fig 7D – can't see anything where the yellow arrow is pointing

Line 410 – include words "CSF1R inhibitor-treated", e.g. throughout the adult life of CSF1R inhibitor-treated wild-type and 5xFAD...

Line 539 – add words "currently available" (method capable of achieving sustained...)

Line 869 – text says 1d and 3d PLX5622, Fig 2D shows 3d and 5d of dosing

Supp Scheme 1 – scheme shows iPrMgBr, protocol (steps 4 and 5) says iPrMgCl, both for the conversion of 5 to 6 and 6 to 8

The specific details of the in vivo AD model are outside my area of expertise, and will defer to a full assessment from other reviewers.

Reviewer #3 (Remarks to the Author):

In this manuscript, Spangenberg et al. demonstrated that microglia facilitate plaque formation and compaction in a transgenic AD mouse model. In the absence of microglia, Abeta deposits in cortical blood vessels with sporadic plaques forming close to the few surviving microglia. Gene expression profiling of brain homogenate with few remaining microglia showed a DAM profile. Depleting microglia in 5xFAD mice prevented alterations in the expression of inflammatory genes and synaptic genes. These are interesting observations; however, the manuscript lacks proper interpretation of behavioral tests as well as validation of the gene expression analysis. Although, the finding that microglia facilitate plaque formation is plausible, previous publication from the same group showed beneficial effect of microglia depletion on behavioral improvement in AD mouse models (PMID: 26921617). Moreover, previous report from different group showed reduced plaque load by early long-term administration of PLX3397 in 5XFAD mice (PMID: 29490706),

which was also shown in this manuscript. Despite these similarly reported data, the novelty of this work is that repopulation of microglia results in a reappearance of Abeta plaques, which is important for the field and understanding the contribution of microglia to AD pathogenesis.

Major comments:

1. Figure 3D: the authors showed that other leukocyte subsets were not altered by PLX treatment. However, their n-size is small and two of the markers tested were very close to significance (CD11b and CD19). The authors should increase the group size and show the circulating leukocyte subsets for the 5xFAD mice groups as well.
2. Figure 3G-K: the interpretation for behavioral tests is very confusing. The authors concluded that the long-term absence of microglia is not detrimental to murine cognitive function and may be beneficial, because microglia-depleted WT mice performed better than the untreated mice. However, the authors also described that "Compared to wild-type animals, 5xFAD mice showed altered behaviors in the elevated plus maze, and these behaviors were further exacerbated with microglial elimination (Fig. 3G, H)."
3. Figure 8: The alteration of synaptic genes expression should be validated. The morphology and function of the synapses with PLX-treatment should be shown.
4. Figure 6A, B: The authors should discuss why there are surviving microglia in the subiculum of treated 5xFAD mice but not in treated WT mice. One explanation might be that the expression of CSF1R is downregulated in the plaque-associated microglia.
5. Figure 10: There are no behavioral tests or gene expression analysis with the mice in the microglia repopulation groups.
6. The finding that amyloid accumulates in the blood vessels in PLX-fed mice is intriguing. It would be interesting to know whether these deposits in the vasculature have any functional consequences i.e. do they impact the integrity of the BBB barrier?
7. Previous studies from the authors' group show that chronic elimination of microglia could improve the cognitive function in 5xFAD mice. However, there was no obvious improvement in behavioral tests in the PLX-treated AD mice in this manuscript. This should be discussed.
8. Figure 5K: The expression of BACE1, PS1 is increased in PLX-treated WT mice as compared to untreated ones. The authors should discuss the results.

Minor comments:

9. Figure 6I, J: Why there are less GFAP+ astrocytes in treated 5xFAD mice? Maybe related to glia-glia communication. The authors should discuss this.
10. Figure 1: why do the authors use 3xTg mice to test the function of non-plaque associated microglia, and not in 5xFAD mice?

Dear Editor and Reviewers,

We thank you for your comments and we are excited to resubmit our research article “Synthesis of a specific CSF1R inhibitor for sustained microglial depletion reveals crucial roles of microglia in plaque development and transcriptional alterations in Alzheimer’s disease mice.” To address the comments, we have added supporting data and revised the text. We hope that you find the amended manuscript suitable for publication. We have colored all text changes in red in the manuscript and detailed our responses to the specific comments below.

In addition we have made improvements to the RNA-seq website, and now display adjusted p-values, as well as RPKM values, to help people explore how 6-months of microglial depletion combined with AD pathology modulates gene expression across 3 brains regions (http://rnaseq.mind.uci.edu/green/ad_plx/gene_search.php).

Sincerely,

Kim Green

Referee: 1

Comments to the Author

It has been reported previously that the acute ablation of microglia triggers ataxia-like behavior. “Microglia-depleted mice progressively developed severe motor coordination dysfunction similar to spinocerebellar ataxia, including kyphosis, loss of balance and hind limb claspings.” (Rubino, ..., Butovsky, Lassmann & Weiner Nature Communications volume 9, Article number: 4578 (2018)). These phenotypes would likely influence the behavioral assays the authors performed in this study. Therefore, the authors should test whether depletion of microglia with PLX5622 causes any of the phenotypes described above and determine whether their ablation system also affects locomotion as measured by rotarod testing.

We thank reviewer #1 for this critique. While we did not employ any specific behavioral assays to evaluate ataxia-like behavioral deficits in this study, we did not detect any differences in distance traveled or average speed between groups during either Elevated plus maze or Morris water maze, suggesting that motor coordination is not overtly impaired, as shown below.

In Rubino et al., ataxia-like deficits manifest following a combination of microglial depletion and subsequent repopulation, as IBA1⁺ cell number is elevated relative to controls at the time of behavioral assessment. Moreover, they utilize diphtheria toxin driven by the CX3CR1 promoter to deplete microglia, which is reported to induce a potentially neurotoxic cytokine storm (Bruttger et al., 2015; elevated expression of TNF- α , IL-1 β , CXCL9, 10, and CCL2, 5, and 7, for example). In fact, Rubino et al., in the same study found that microglial depletion via CSF1R inhibitor treatment in diphtheria toxin-treated or control mice prevented these ataxia-like behaviors, i.e. “Control mice fed PLX5622 diet did not demonstrate signs of ataxia or exhibit upregulation of type 1 interferon responsive genes”. As such, it appears that the ataxia-like phenotypes are caused by a cytokine storm resulting from diphtheria toxin-induced myeloid cell death that can be prevented with PLX5622 (Rubino et al., 2018) and which is not observed in our model of microglial depletion (Elmore et al., 2014). Thus, CSF1R inhibitors are currently the only viable approach to eliminate microglia in a specific, non-invasive, and sustained fashion. For these reasons, our current study illustrating for the first time the structure and synthesis pathway for PLX5622, combined with *in vivo* data showing that we can eliminate microglia for at least 6 months, even in models of aggressive AD pathology, is important to advance our understanding of microglial biology in AD.

• Line 280. The authors have to comment on the why there is such a huge difference in green signal (IBA1 staining) in the Wildtype+PLX5622 versus 5xfAD+PLX5622. We expect that these two conditions would have almost no green signal, especially from their statistical analysis, but instead the Wildtype looks+PLX5622 looks high in green signal (Figure 3C). As the manuscript stands, the small inset appears to be selected with the bias towards of giving the lowest possible green signal value in Wildtype+PLX5622.

We thank reviewer #1 for this comment. We agree that there is a difference in background intensity signal for the whole hemisphere stiches showing IBA1 immunoreactivity between WT+PLX5622 and 5xfAD+PLX5622. However, this is not uncommon in immunohistochemical assays, particularly when the antigen of interest is expressed in low levels (i.e., immunolabeling for IBA1⁺ microglia in a microglia-depleted mouse). To exclude any effect of background signal intensity,

we analyzed IBA1⁺ cell number using the spots module in Imaris software, which counts specifically circular shapes of a given size and fluorescent label, thereby normalizing any effect of background.

• Line 337: “The absence of microglia did not significantly alter the levels of A β 38, A β 40, or A β 42 in detergent-soluble or -insoluble cortical and thalamic homogenates in either the 4- or 7- month cohorts (Fig. 5A-H).” This is not an accurate statement. There is a significant reduction in 5B. In Fig 5A and C there are trends in this direction as well. When looking at the individual data points, the lack of difference is likely a matter of too few data points. In some cases, only 3 mice are used (Fig 5D) and this will bias the data to lack of significant difference. In all figures of the paper pertaining to histology and other kinds of protein quantification, the authors should use at least 7 mice (or data points) in order to perform a meaningful statistical analysis. This is also obvious in figure 5L where signal intensity of BACE1 or PS-1 could become significant, if more than 4 data points were collected. Looking at the blot in figure 5K, C-term PS1 looks to be increased by PLX5622 treatment in Wildtype mice, but a statistical analysis with such few data points will not reflect this. We agree with reviewer #1 and have added additional *n* (7-8/group) to the biochemical analyses. With the increased *n*, we find no differences between soluble and insoluble A β 38, A β 40, or A β 42 with microglial depletion. For western blot analyses, we also find that C-terminal APP fragments C99 and C83 are unchanged in 5xfAD treated with PLX5622 with the increase in *n*. The figure has been updated accordingly.

Minor points:

• Line 141, the authors should specify the kind of fixation used (drop-fixed as in line 91, or intracardial PFA?).

We thank the reviewer for their comment and have updated the text accordingly.

• Line 233. The authors describe the accumulation of Thio-S+ material in non-plaque associated microglia. One drawback about thioflavin S staining is the lack of specificity as it non-selectively binds beta sheet contents of proteins. The authors should confirm that non-plaque associated microglia indeed contain intracellular β -amyloid aggregates using an antibody detecting β -amyloid.

We thank the reviewer for their comment and have added images of brain sections immunolabeled for 6e10, which confirm our prior results using Thio-S (Figure 1M, P, S). We have also confirmed these findings using Amylo-Glo, another known stain for fibrillar A β .

• Line 292. The authors claim that long-term absence of microglia is not detrimental to murine cognitive function. It is possible that the performance in the MWM test is not affected but other tests would reveal a difference. The authors may consider to qualify this statement or to perform additional behavioral tests.

We have revised the text accordingly. We originally performed a battery of behavioral tasks including Contextual fear conditioning and Spontaneous alternation Y-Maze (Supplementary Figure 1), but due to space limitations, did not include these data in the original submission. We have added the additional behavior to the supplementary document, demonstrating that extended microglial elimination does not induce cognitive deficits in these tasks, as well. In a previous study, we eliminated microglia for 8 weeks and saw no deficits in Contextual fear conditioning or Barnes maze, providing further support that extended durations of microglia-depletion are not determinantal to cognitive functioning in mice (Elmore et al., 2014). We have added text to the manuscript to clarify this point.

• Line 342: “Investigations into... found no significant changes... (Fig 5I,J).” From the figure 5I it looks like the A11 intensity is much weaker in 5XFAD+PLX5622 than in untreated 5XFAD mice. Perhaps a more representative image would be preferable.

We agree with the reviewer’s comment and have included a more representative image of A11 immunoblotting in the revised manuscript.

• Line 378” ... and lacked the homogenous staining intensity of plaques in microglia-intact 5xfAD mice.” The authors need to clarify exactly how staining intensity was quantified in Fig 6I. From the micrographs, the green signal intensity looks virtually identical, at least in the center or the plaques (Figure 6F1 and 6G1).

We apologize for the lack of clarity. We agree that the center of the plaque looks equally intense between microglia -depleted and -intact mice. However, in our quantification, we measured Thio-S intensity over the entire plaque, including the fibrils radiating out from the plaque center. This allowed us to detect differences in plaque Thio-S intensity averaged over the entire plaque and is a quantitative method previously established by Condello et al., 2015, who originally identified and defined methods to quantify plaque compaction. These methods have since been validated by additional groups (Pahzhikar et al., 2019; Yuan et al., 2016).

• Line 395 “Treated 1.5-month-old 5xfAD mice with this formulation until 5 months of age” versus line 398 “Treating 1.5-

month-old 5xfAD mice for 5 months”. This is probably a typo, but the authors should clarify if the mice are 5 months old or treated for 5 months with these drugs.

We apologize for the oversight – animals were treated until 5 months of age. The text has been changed accordingly.

- Figure 3D-K. The authors should use consistent data display design, rather than combining bars and violin plots. The chosen colors (gray and purple shades) of the different groups are also not easily discernible. The data would be clearer with colors like black, white, red, blue, green and so on. There is also no reason to use different symbols for the different bars since these are clearly separated by distance. In order to visualize the individual data points in a clear manner, the authors should consider using open circles in all cases with a thin border that clearly contrasts in color to the bars.

We agree with reviewer #1 and have updated the color scheme for better readability.

- The legends of the figures describe n’s used as a range, e.g. line 908 “n=8-12 for Wildtype” etc. Exact numbers should be given in all cases.

We thank reviewer #1 for this comment and agree with their concern. However, to reduce redundancy, we chose to use a range in the figure legends, as groups (e.g., Wild-type / PLX5622 / 5xfAD / 5xfAD + PLX5622) within a bar graph display the n as individual replicates superimposed over each bar.

- Line 408: “Gene expression analyses of Wild-type and 5xfAD mice in the presence and absence of microglia, across brain regions” The authors should clarify if the mice were ablated with PLX3397 or PLX5622.

We appreciate the reviewer’s comment and have updated the text accordingly.

- Line 416. “extracted,and...” space is missing

We appreciate the reviewer’s comment and have updated the text accordingly.

- Line 575: “... small subset of surviving CSF1R-resistant microglia”. Perhaps a better wording would be “... small subset of surviving microglia resistant to CSF1R-inhibition”

We appreciate the reviewer’s comment and have updated the text accordingly.

- Fig10J: Because there are such large differences between wildtype and 5xfAD in “# of GFAP+ Cells / FOV”, the authors should consider splitting the graph, so that the reader can better see the data points of the wildtype mice in the three conditions. The authors should also comment on the increase in number of GFAP positive cell sin the repopulating wildtype mice. Is this something observed in other brain regions? The same analysis should be performed for “RS cortex” and “Thalamus” for consistency with previous figures.

We agree with reviewer #1 and have updated Figure 10J accordingly. The increase in GFAP immunoreactivity coincides with the reappearance of both microglia and plaques in the brain. Thus, we do not see any significant changes in GFAP⁺ in microglia-repopulated wild-type animals. We have included these data in the supplementary document. In addition, we added astrocyte analyses from the 6-month treated group of mice and show results for multiple brain regions (Supplementary Figure 3). Because astrocytes in the hippocampus and thalamus exhibit basal expression of GFAP, we focused our analyses on the cortex, as astrocytes only express GFAP in this region when they become reactive (i.e. when associated with a plaque). Thus, evaluating GFAP expression in the cortex allows us to determine alterations in astrocyte reactivity.

Reviewer #2 (Remarks to the Author):

Thank you for the opportunity to review Nature Communications Manuscript#: NCOMMS-19-01388-T.

Overall the manuscript is well-written and presents interesting scientific hypotheses that are backed by data. The authors demonstrate the complete, long-term elimination of microglia in WT and 5xfAD mice, which is accomplished by chow administration of selective and brain penetrant CSF1R inhibitor PLX5622. This level and length of microglial depletion is unprecedented, and enables the detailed evaluation of changes to AD-associated pathology, including brain plaques and gene expression. This data is used to generate hypotheses about the potential causative role of microglia in the progression of AD pathology based on this mouse model data. The results are of immediate interest for the neuroscience community, and the quality of data is excellent.

There are a few conclusions needing attention.

(1) In general, although the association of microglial depletion in the brain with findings is plausible, in many instances the absence of microglia is not proven to be definitively causative of observed effects. The manuscript should be re-worded to

that effect. Also the findings are in a mouse model, not in AD itself and that should be clarified. Some examples with suggested updates are given:

- Line 310: “suggest that microglia may be critical regulators”
- Line 382: “some threshold density of microglial cells may be required”
- Line 386: “these results are consistent with prior studies”
- Line 446: “Microglial depletion results in the downregulation of”
- Line 465: reword as “in the absence of microglia, AD-pathology induced down regulation of neuron-associated genes was prevented”
- Line 467: reword as “microglia in the 5xfAD brain may drive downregulation”
- Line 515: remove word “directly”
- Line 552: ideal is overly strong - suggest “a useful tool compound for investigating”
- Line 561: “myeloid biology appears to be a large contributor”
- Line 610: “our study suggests that the presence”
- Line 627: “the absence of microglia is associated with no alterations”
- Line 628: replace demonstrating with suggesting
- Line 629: “and beneficial roles in a preclinical model of AD”
- Line 636: “microglial elimination is associated with prevention of formation of plaques”
- Line 637: “that occur in a preclinical model of AD progression”
- Line 638: “microglia appear to contribute to the multiple facets”

We thank reviewer #2 for their comments and have made the above changes to the manuscript.

(2) There are instances where figure information and/or statistics do not agree with what is stated in the manuscript text.

- Line 320: “across all brain regions (Fig. 4N)” – figure shows NS in the cortex

We agree with the reviewer’s concern and have changed the text from “all” to “many”.

- Line 351: “no overt changes” (Fig 5M) – however significant changes observed in *Inpp5d*, *Spi1*, *Trem2*, not mentioned in text

Thank you for your comment. We have changed the text to “and we found minimal changes with microglia depletion, other than in microglial expressed genes (i.e. *Trem2*, *Spi1*, *Inpp5d*, and *Ctsd*).”

- Lines 437-442: only 3 genes (*Ccl6*, *Clec7a*, *Cst7*) appear to be present in the hippocampus but not cortex or thalamus in the PLX5622-treated 5xfAD animals – “these genes” implies that all 6 genes are present, as well as the other ~14 genes listed in the text. Please clarify wording.

Thank you. We changed the text to “we detected expression of many of these genes (i.e. *Ccl6*, *Clec7a*, and *Cst7*) in PLX5622-treated 5xfAD hippocampi”. However, if you explore the gene expression at http://rnaseq.mind.uci.edu/green/ad_plx/gene_search.php, you can see that many of these genes exhibit elevated expression only in the hippocampus (i.e., comparing Wild-type+PLX5622 with 5xfAD+PLX5622).

- Line 449, 453, 462 – I count 413 genes (not 414) in Fig 9A

We apologize for the oversight. The text has been changed accordingly.

- Figure 10E – this appears to be 5xfAD + PLX5622 but is labeled as 5xfAD + Repopulation

We thank you for the comment. Figure 10E is labeled correctly as 5xfAD + Repopulation. These images are demonstrating the association of repopulated microglia with parenchymal deposits and the continued presence of CAA in microglia-repopulated 5xfAD mice.

(3) The structural explanation for the observed improved broad kinome selectivity of PLX5622 is unpersuasive, and no rationale for the improved brain penetration of PLX5622 is provided.

- Lines 252-255 and 861-863: A fluorine atom has approximately the same atomic radius as a hydrogen atom. The steric argument that a bulkier cysteine in KIT and FLT3 is the rationale for greater selectivity does not make sense.

Thank you for the comment. The van der Waals radius of F atom (1.35-1.47Å) is slightly greater than that of H atom (1.1-1.2Å). The bond length of C-F is 1.37Å, longer than that of C-H (1.09Å). The combined effect is that the van der Waals edge of F is extended by about 0.5 Å compared to H. While this difference may seem small, the effect appears to be significant here. Anchored on both ends by hydrogen bonds and hydrophobic packing (Fig. 2B), PLX5622 has the fluorine atom at a fixed position. As van der Waals repulsion increases in proportion to the 6th power of the distance between two atoms, a larger gate keeper residue (as seen in KIT and FLT3) will likely incur an energetic penalty for the steric

hindrance. In support of this mechanism, a close analogue of PLX3397 containing the same 2-fluoro substitution as PLX5622 has the same CSF1R potency as PLX3397 but is 10-fold more selective over KIT than PLX3397. This has been incorporated into the revised text.

- Lines 255-259 and 865-866: is there x-ray data to support the stabilization of the allosteric pocket of CSF1R? If not, then need to specify that this is a hypothesis.

This is indeed a hypothesis. “Stabilize” has been changed to “occupy”.

- Figure 2B shows an H-bond interaction of the pyridine N with the protein backbone but this is not mentioned in the text or caption. The key hinge-binding interactions are also not described.

The legend for Figure 2B has been revised to include descriptions on hydrogen bond interactions.

- Lines 260-262: what physicochemical characteristics of PLX5622 are responsible for the increased brain penetration? Was this predicted?

The optimization of brain penetration was guided by a general understanding of how physicochemical properties of a compound might influence its ability to cross the blood-brain barrier (BBB). For example, compounds with lower molecular weight, higher lipophilicity, and higher permeability trend towards better BBB penetration. Although physicochemical properties did prove to be a useful guide, PLX5622 was selected based on experimentally measured BBB partition in rodents (rats or mice). In retrospect, the improved BBB penetrance of PLX5622 over PLX3397 is somewhat correlated with their physicochemical properties. PLX5622 has lower molecular weight, higher lipophilicity (as measured by LogD) and better Caco-2 cell permeability. We have included the data in the new supplementary Table 6 and described the findings in the text.

(4) It is unclear how testing the less selective (and structurally highly similar) reagent PLX3397 rules out “potential confounding off-target effects.” (Lines 392-393).

- Considering the cellular activity (THP1 = 0.032 μ M), why at a concentration of \sim 1 μ M in the brain is no effect on microglia observed? What is the unbound brain concentration & is there high brain protein binding?

We agree with reviewer #2 and have removed “potential confounding off-target effects from the text”. At a concentration of 1 μ M, PLX3397 is likely modulating microglial phenotype but is insufficient to induce microglial death.

- Line 596 – what do you mean by “off target CSF1R inhibition”? Suggest striking this.

We apologize for the lack of clarity. We will remove “off target” effect from the text.

(5) Parts of the discussion section are confusing.

- Lines 578 -586 - what data suggests an interplay between ApoE and microglia facilitating plaque pathogenesis, other than a correlation in the timing?

We have clarified the ApoE discussion further in the manuscript and include an additional figure to demonstrate our point (Fig. 6J-L). We immunolabeled 7-month-old microglia-intact and -depleted animals for ApoE and find a strongly diminished ApoE signal in both microglia and plaques with PLX5622 treatment, and instead, ApoE accumulates within Thio-S positive blood vessels. Because this phenotype is reminiscent of the pathology observed in animal models that modulate ApoE expression (Ulrich et al., 2018; Holtzmann et al., 2000; Bales et al., 2000; Fryer et al., 2003), we believe there is a convergence of the two pathways (i.e., microglia and ApoE) on neuropathology in AD. The discussion has been updated accordingly to clarify these points.

- Lines 602: include a cross reference for the 2nd model of AD (Fig 1, A-F), and for the human brain (Fig 1, K-O).

We apologize for the lack of clarity and have added the above figure references to the discussion.

Previous literature is referenced appropriately, with two suggestions.

1. A recent review of AD advancements with regard to microglia would be appropriate to include, e.g. Brit. J. Pharm. 2018, “The involvement of microglia in Alzheimer’s disease: a new dog in the fight.” DOI:10.1111/bph.14546

2. Description of the phenotype of CSF1R-deficient mice and humans with CSF1R mutations would enhance the manuscript.

- a. E.g. (mice) PLoS One 2011; “Absence of colony stimulation factor-1 receptor results in loss of microglia, disrupted brain development and olfactory deficits.” DOI:10.1371/journal.pone.0026317.

b. There are multiple examples of CSF1R mutations in humans that results in hereditary diffuse leukoencephalopathy with spheroids; this example focuses on impact on microglia. Annals of Neurology 2016. "Characteristic microglial features in patients with hereditary diffuse leukoencephalopathy with spheroids." DOI:10.1002/ana.24754

We thank reviewer #2 for their thorough literature search on the subject and have included the above manuscripts in the discussion. I don't know if you saw it but a paper came out last month detailing a human with a homozygous CSF1R mutation – the child lived for 4 months, but upon examination was found to have no microglia (<https://www.sciencedirect.com/science/article/pii/S0002929719301065?via%3Dihub>)!

A few additional recommendations, for clarity:

Line 32 – mice (not animals)

Line 45 – in the AD brain of rodents (otherwise may imply human, and minimal human data is shown)

Line 74 – of multiple facets of AD pathology (since this is a model, and not the actual disease)

Line 97/98 – are the specific protease and phosphatase inhibitors obvious to those skilled in the art?

Lines 295/296, also line 588 – no evidence of "shuttling" of A β is provided, simply the finding that A β is present in the vasculature. Could there be sequestering of A β ? Or some other interpretation? This should be re-worded for accuracy.

The discussion in lines 605-608 are reasonable.

Lines 332, 336 – "Data not shown." Minimally should include this data in supporting information.

Line 397 – Fig 7D – can't see anything where the yellow arrow is pointing

Line 410 – include words "CSF1R inhibitor-treated", e.g. throughout the adult life of CSF1R inhibitor-treated wild-type and 5xFAD...

Line 539 – add words "currently available" (method capable of achieving sustained...)

Line 869 – text says 1d and 3d PLX5622, Fig 2D shows 3d and 5d of dosing

Supp Scheme 1 – scheme shows iPrMgBr, protocol (steps 4 and 5) says iPrMgCl, both for the conversion of 5 to 6 and 6 to 8

We thank you for the above comments and have changed/clarified the text accordingly.

The specific details of the in vivo AD model are outside my area of expertise, and will defer to a full assessment from other reviewers.

Reviewer #3 (Remarks to the Author):

In this manuscript, Spangenberg et al. demonstrated that microglia facilitate plaque formation and compaction in a transgenic AD mouse model. In the absence of microglia, Abeta deposits in cortical blood vessels with sporadic plaques forming close to the few surviving microglia. Gene expression profiling of brain homogenate with few remaining microglia showed a DAM profile. Depleting microglia in 5xFAD mice prevented alterations in the expression of inflammatory genes and synaptic genes. These are interesting observations; however, the manuscript lacks proper interpretation of behavioral tests as well as validation of the gene expression analysis. Although, the finding that microglia facilitate plaque formation is plausible, previous publication from the same group showed beneficial effect of microglia depletion on behavioral improvement in AD mouse models (PMID: 26921617). Moreover, previous report from different group showed reduced plaque load by early long-term administration.

of PLX3397 in 5XFAD mice (PMID: 29490706), which was also shown in this manuscript. Despite these similarly reported data, the novelty of this work is that repopulation of microglia results in a reappearance of Abeta plaques, which is important for the field and understanding the contribution of microglia to AD pathogenesis.

[REDACTED]

Major comments:

1. Figure 3D: the authors showed that other leukocyte subsets were not altered by PLX treatment. However, their n-size is small and two of the markers tested were very close to significance (CD11b and CD19). The authors should increase the group size and show the circulating leukocyte subsets for the 5xFAD mice groups as well.

We thank reviewer #3 for their comment. As the study requires six months of treatment to generate additional animals for the flow cytometry analysis, we cannot easily add more animals to analyze/characterize leukocyte subsets. If the reviewer agrees, we will remove these data from the manuscript, although the n=5 is sufficient to see the trends (which are noted in

the figure).

2. Figure 3G-K: the interpretation for behavioral tests is very confusing. The authors concluded that the long-term absence of microglia is not detrimental to murine cognitive function and may be beneficial, because microglia-depleted WT mice performed better than the untreated mice. However, the authors also described that “Compared to wild-type animals, 5xFAD mice showed altered behaviors in the elevated plus maze, and these behaviors were further exacerbated with microglial elimination (Fig. 3G, H).”

We thank reviewer #3 for their comment. The comment about long-term absence of microglia not being detrimental to cognitive functioning, and even showing benefits, was specifically directed at data from the wild-type animals, which is consistent with our prior studies and data published from other groups using CSF1R inhibitors.

Microglia play different roles in disease states than in non-diseased brains, and their elimination in diseased brains can help dissect out their roles in any behavioral impairments. There are now many examples where microglia have either beneficial (Rice et al., 2015; Szalay et al., 2016) or detrimental (Rice et al., 2015; Spangenberg et al., 2016; Acharya et al., 2016) roles in behavior in disease states, and appears dependent on the nature of the disease and the disease state. Here, the most striking behavioral phenotype that we observe in this age of 5xFAD mice is the elevated plus maze, which is employed to evaluate animal anxiety. We have clarified this point in the text and highlight that this behavior in 5xFAD mice is further exacerbated with microglia elimination, suggesting that microglia have beneficial roles in this particular task at this stage of the disease in these mice. Importantly, 5xFAD mice at this age (7 months) do not have robust cognitive impairments, and so there were no deficits to recover with microglial elimination (for example, in the Morris Water Maze). Of note, our prior study, which eliminated microglia in aged 5xFAD mice with existing extensive pathology, were 11 months of age and had cognitive deficits that were then reversed (Spangenberg et al., 2016).

I would point out that we are part of the MODEL-AD consortium (<https://model-ad.org/>) and have been robustly phenotyping 5xFAD mice across their lifespans, and confirm that 5xFAD mice do not show cognitive deficits until advanced ages, but show elevated plus maze changes from an early age. These results have been replicated across the MODEL-AD consortium (UCI, Indiana University, and the Jackson Labs), and the data will be available soon via the synapse website (<https://www.synapse.org/>).

3. Figure 8: The alteration of synaptic genes expression should be validated. The morphology and function of the synapses with PLX-treatment should be shown.

As validation of our hippocampal RNA-seq findings, we utilized a NanoString nCounter Gene Expression Assay (Kulkarni et al., 2011), which digitally detects mRNA molecules of interest using target-specific probe pairs without requiring the conversion of mRNA to cDNA by reverse transcription or amplification of cDNA by PCR. Thus, NanoString analyses provide a highly sensitive readout of gene expression. We confirmed that expression of disease-associated microglial genes (*Itgax*, *Trem2*, *Grh*, *Tgfb1*, *Gclrs*, etc.) and homeostatic microglial genes (*P2ry12*, *Tmem119*, *Csf1r*, *Cx3cr1*, etc.) are increased in 5xFAD mice relative to wild-type animals. Moreover, neuronal genes including *Scna*, *Gls*, *Apc*, *Dlgap1*, *Nrxn1*, etc. are downregulated in 5xFAD animals compared to wild-type and this effect is normalized (i.e., the expression of neuronal genes in 5xFAD mice returns to wild-type levels) in microglia-depleted 5xFAD animals.

As stated above, the study requires six months of treatment to generate additional animals for morphological (e.g., electron microscopy and/or Golgi-cox staining) or functional (long-term potentiation) analyses of hippocampal synapses. While we agree these data would strengthen the manuscript, we cannot easily add more animals to analyze/characterize synaptic endpoints. However, we previously reported that one month of microglia-depletion in 5xFAD mice reduced overall dendritic spine density in the CA1 and the elimination of microglia consequently prevented this loss in dendritic spines (Spangenberg et al, 2016), and have added this to the discussion.

4. Figure 6A, B: The authors should discuss why there are surviving microglia in the subiculum of treated 5xFAD mice but not in treated WT mice. One explanation might be that the expression of CSF1R is downregulated in the plaque-associated microglia.

We thank reviewer #3 for the comment. In 5xFAD mice, the subiculum is one of the first areas to develop pathology (Oakley et al., 2006). Thus, we believe the population of surviving microglia may be due to pathology-induced changes in this region that allow microglia to persist/survive with CSF1R inhibitor treatment. This could be due to an upregulation in *Trem2*, which acts as a survival signal for microglia in response to A β plaques when CSF1R signaling is inhibited (Wang et al., 2015). In line with this, our internal RNA-seq database shows that CSF1R transcripts are almost completely absent in PLX5622-treated 5xFAD mice in the thalamus and cortex, but the hippocampus shows a low expression level of CSF1R. This suggests that the remaining plaque-associated microglia in the subiculum are the predominant cellular source of CSF1R expression, and that these cells survived CSF1R inhibitor treatment due to reasons other than a downregulation of CSF1R.

5. Figure 10: There are no behavioral tests or gene expression analysis with the mice in the microglia repopulation groups.

We thank reviewer #3 for the comment. As mentioned before, 5xfAD mice at five months of age (i.e., when the experiment was concluded) do not show cognitive impairments. Therefore, we did not pursue behavioral testing for this particular experiment. As such, the repopulation study was performed only to determine the influence of returning microglia on plaque pathology (i.e., whether the reappearance of microglia would result in plaque formation).

6. The finding that amyloid accumulates in the blood vessels in PLX-fed mice is intriguing. It would be interesting to know whether these deposits in the vasculature have any functional consequences i.e. do they impact the integrity of the BBB barrier?

We thank reviewer #3 for their comment. We performed Prussian blue staining of animals in the 4- and 7-month-old cohorts and find evidence of microhemorrhages (i.e., iron reactivity) in the thalamus of two microglia-depleted 5xfAD animals in the 4-month old mice. However, the averaged result of the iron staining of each animal revealed no significant differences between groups at the 4-month time point. By 7-months of age, we find minimal iron reactivity in the thalamus of microglia-depleted 5xfAD animals, but as with the 4-month cohort, this effect is not significant compared to other groups. We included these images in Supplementary Figure 5. Moreover, we immunolabeled 4-month-old tissue for the endothelial marker claudin-5, revealing a preferential deposition of A β with larger diameter Claudin-5⁺ blood vessels and a reduction in Claudin-5 intensity in blood vessel regions exhibiting A β accumulation. We have included these data in the supplementary figure, as well.

7. Previous studies from the authors' group show that chronic elimination of microglia could improve the cognitive function in 5xfAD mice. However, there was no obvious improvement in behavioral tests in the PLX-treated AD mice in this manuscript. This should be discussed.

Please see comment above (and pasted here):

Importantly, 5xfAD mice at this age (7 months) do not have cognitive impairments, and so there were no deficits to recover with microglial elimination. Of note, our prior study, which eliminated microglia in aged 5xfAD mice with existing extensive pathology, were 11 months of age and had cognitive deficits that were then reversed (Spangenberg et al., 2016).

I would point out that I am part of the MODEL-AD consortium (<https://model-ad.org/>) and have been robustly phenotyping 5xfAD mice across their lifespans, and confirm that 5xfAD mice do not show cognitive deficits until advanced ages, but show elevated plus maze changes from an early age. These results have been replicated across the MODEL-AD consortium (UCI, Indiana University, and the Jackson Labs), and the data will be available soon via the synapse website (<https://www.synapse.org/>).

We have added these points to the discussion.

8. Figure 5K: The expression of BACE1, PS1 is increased in PLX-treated WT mice as compared to untreated ones. The authors should discuss the results.

While BACE1 and PS1 do increase in WT animals treated with PLX5622, the overall effect was not significant (or trending) which is why we decided to leave this out of the discussion. Additionally, we added additional *n* per Reviewer #2's request and now find no differences between WT and PLX5622 animals (trending or significant).

Minor comments:

9. Figure 6I, J: Why there are less GFAP+ astrocytes in treated 5xfAD mice? Maybe related to glia-glia communication. The authors should discuss this.

We thank reviewer #3 for their comments. Microglia are upstream regulators of astrocyte activity (Pascual et al., 2012) and induce a reactive astrocyte subtype present in AD (Liddel et al., 2017). Therefore, our findings of reduced astrocyte reactivity (i.e., GFAP expression) in the absence of microglia are in support of these reports, and provide additional evidence that microglia regulate the astrocyte response in AD. We have added these points to the text for clarification.

10. Figure 1: why do the authors use 3xTg mice to test the function of non-plaque associated microglia, and not in 5xfAD mice?

We thank reviewer #3 for the comment. We included data from 3xTg-AD mice showing plaque-distal microglia containing A β aggregates to demonstrate that this phenotype is observed in multiple mouse models of AD, in addition to the human brain itself (Fig. K-P) and 5xfAD mice (shown in Figure 1G-J) used for the rest of the study, lending strength to the

argument that this phenotype is real and conserved among different AD models and species.

REVIEWERS' COMMENTS:

Reviewer #1 (Remarks to the Author):

The authors have sufficiently answered all points raised by this reviewer. I have a couple of suggestions for improving the general data presentation before publication.

In Figure 9 the authors should make clear what software and parameters (cut-offs, numbers and p-values etc.) were used for the GO analysis, and this information should be included in the "methods".

Whenever the authors indicate statistical significance, they should do so not only with asterisks and horizontal lines, but also with vertical lines (one per bar), to help the reader understand which differences are significant. One example is figure 10F. Here, vertical lines that clearly indicate which bars are being compared will help the reader, especially in the thalamus data, where the bars differ greatly in size. The authors should add this whenever more than two bars are compared in one figure.

Reviewer #2 (Remarks to the Author):

Thank you for the opportunity to review revised Nature Communications manuscript #: NCOMMS-19-01388-T. The authors have sufficiently addressed previous feedback, with two minor comments remaining.

Supporting Scheme 1 and Protocol steps 4 and 5 still do not agree. The scheme (for conversion of 5 to 6 and of 6 to 8) shows the use of the bromo Grignard reagent $i\text{PrMgBr}$ while the protole (in both steps 4 and 5) cites use of the chloro Grignard reagent $i\text{PrMgCl}$. Please clarify whether the bromo or chloro reagent was used.

The final reference (#77) says only "!!! INVALID CITATION !!!"

Reviewer #3 (Remarks to the Author):

The authors have addressed all the Reviewer's comments. In regards to the comment #1 (Figure 3D), the authors are not required to remove the data (n=5).

REVIEWERS' COMMENTS:

Reviewer #1 (Remarks to the Author):

The authors have sufficiently answered all points raised by this reviewer. I have a couple of suggestions for improving the general data presentation before publication.

In Figure 9 the authors should make clear what software and parameters (cut-offs, numbers and p-values etc.) were used for the GO analysis, and this information should be included in the "methods".

We have included a description of the GO analysis in the Methods, including the package used for analysis as well as the statistical method used (i.e., Benjamini and Hochberg) and types of p-values generated from the statistical test (cut off at $p < 0.10$).

Whenever the authors indicate statistical significance, they should do so not only with asterisks and horizontal lines, but also with vertical lines (one per bar), to help the reader understand which differences are significant. One example is figure 10F. Here, vertical lines that clearly indicate which bars are being compared will help the reader, especially in the thalamus data, where the bars differ greatly in size. The authors should add this whenever more than two bars are compared in one figure.

We added vertical lines to all bar graphs that display more than two bars.

Reviewer #2 (Remarks to the Author):

Thank you for the opportunity to review revised Nature Communications manuscript #: NCOMMS-19-01388-T. The authors have sufficiently addressed previous feedback, with two minor comments remaining.

Supporting Scheme 1 and Protocol steps 4 and 5 still do not agree. The scheme (for conversion of 5 to 6 and of 6 to 8) shows the use of the bromo Grignard reagent $i\text{PrMgBr}$ while the protocol (in both steps 4 and 5) cites use of the chloro Grignard reagent $i\text{PrMgCl}$. Please clarify whether the bromo or chloro reagent was used.

We apologize for the oversight – both can be used but $i\text{PrMgCl}$ is cheaper. We reviewed the synthesis steps and relabeled the reaction scheme accordingly to just reference isopropylmagnesium chloride.

The final reference (#77) says only "!!! INVALID CITATION !!!"

We thank Reviewer #2 for finding this error in the reference section. Due to reference limits, we decided to remove Reference #77 from the manuscript.

Reviewer #3 (Remarks to the Author):

The authors have addressed all the Reviewer's comments. In regards to the comment #1 (Figure 3D), the authors are not required to remove the data ($n=5$).

We thank Reviewer #3 for their understanding. We will keep Figure 3D in the manuscript.